# Neural circuit mechanisms for transforming learned olfactory valences into wind-oriented movement

**Yoshinori Aso[1]\*, Daichi Yamada[2], Daniel Bushey[1], Karen L Hibbard[1], Megan Sammons[1], Hideo Otsuna[1], Yichun Shuai[1], Toshihide Hige[2,3,4]\***

[1]Janelia Research Campus, Howard Hughes Medical Institute, Ashburn, United States; [2]Department of Biology, University of North Carolina at Chapel Hill, Chapel Hill, United States; [3]Department of Cell Biology and Physiology, University of North Carolina at Chapel Hill, Chapel Hill, United States; [4]Integrative Program for Biological and Genome Sciences, University of North Carolina at Chapel Hill, Chapel Hill, United States

**\*For correspondence:**
asoy@janelia.hhmi.org (YA);
hige@email.unc.edu (TH)

**Competing interest:** The authors declare that no competing interests exist.

**Abstract** How memories are used by the brain to guide future action is poorly understood. In olfactory associative learning in *Drosophila*, multiple compartments of the mushroom body act in parallel to assign a valence to a stimulus. Here, we show that appetitive memories stored in different compartments induce different levels of upwind locomotion. Using a photoactivation screen of a new collection of split-GAL4 drivers and EM connectomics, we identified a cluster of neurons post-synaptic to the mushroom body output neurons (MBONs) that can trigger robust upwind steering. These UpWind Neurons (UpWiNs) integrate inhibitory and excitatory synaptic inputs from MBONs of appetitive and aversive memory compartments, respectively. After formation of appetitive memory, UpWiNs acquire enhanced response to reward-predicting odors as the response of the inhibitory presynaptic MBON undergoes depression. Blocking UpWiNs impaired appetitive memory and reduced upwind locomotion during retrieval. Photoactivation of UpWiNs also increased the chance of returning to a location where activation was terminated, suggesting an additional role in olfactory navigation. Thus, our results provide insight into how learned abstract valences are gradually transformed into concrete memory-driven actions through divergent and convergent networks, a neuronal architecture that is commonly found in the vertebrate and invertebrate brains.

## Editor's evaluation

This study provides important new insights into how learning affects behavior in the *Drosophila* model. Using a combination of connectomics, neurophysiology, and behavioral analysis, a small group of neurons in the *Drosophila* brain that integrates learned odor valences and promotes odor tracking by driving upwind orientation and movement is described. The study's conclusion is supported by convincing evidence and rigorous quantitative analysis. Insights from the neural circuit mechanism that translates learning-induced plasticity into appropriate behavioral actions will be of broad interest to neuroscientists.

## Introduction

Animals assign a valence to a stimulus based on experience. Such learning events induce an enduring modification in the stimulus-evoked activity of the nervous system and ultimately change the behavioral response to future encounters with the same stimulus. In mammals, the amygdala is the primary

site for valence assignment during Pavlovian learning (*O'Neill et al., 2018*). As a neutral sensory stimulus (conditioned stimulus [CS]) is paired with punishment or reward (unconditioned stimulus [US]), the CS acquires the capacity to evoke valence-specific response patterns in the amygdala (*Grewe et al., 2017*; *Zhang and Li, 2018*). However, the circuit process in which these learning-dependent CS representations lead to concrete motor patterns during memory retrieval is poorly understood. Comprehensive understanding of this process requires detailed knowledge of the downstream connectivity of the plastic CS-representing neurons. Nevertheless, it has been shown that amygdala-dependent valence-specific behaviors are mediated by distinct networks *Gore et al., 2015* whose outputs diverge to different projection areas responsible for aversive or appetitive unconditioned responses (*Beyeler et al., 2016*; *Namburi et al., 2015*). There is also evidence for an alternative mechanism where neurons capable of eliciting opposing behaviors converge on the same target areas or neurons. For example, GABAergic and glutamatergic projection neurons from the lateral hypothalamus to the ventral tegmental areas (VTA) can evoke appetitive and aversive behaviors, respectively. These projection neurons converge on the same population of GABAergic neurons in VTA to differentially control downstream dopaminergic neurons (DANs) (*Nieh et al., 2016*). Thus, both divergent and convergent circuit motifs are considered important for the valence-to-behavior transformation in vertebrates (*Tye, 2018*).

Divergent pathways for valence processing are also evident in the memory circuit in *Drosophila* both anatomically and functionally. In fly olfactory learning, the primary site for CS-US association is the mushroom body (MB), where parallel axon fibers of the odor-encoding Kenyon cells (KCs) are segmented into a series of MB compartments that are defined by the dendrites of MB output neurons (MBONs) and axons of US-encoding DANs (*Aso et al., 2014a*; *Tanaka et al., 2008*; *Figure 1A*). While population activity of KCs represents odor identity (*Campbell et al., 2013*), that of MBONs is less effective in doing so (*Hige et al., 2015b*). Instead, individual MBONs are considered to encode the valence of stimuli because optogenetic activation of each type of MBONs can elicit either approach or avoidance behavior (*Aso et al., 2014b*; *Owald et al., 2015*). However, MBONs do not appear to command specific motor sequences because their activation does not induce stereotyped motor patterns (*Aso et al., 2014b*). Thus, how abstract valence signals carried by MBONs are translated into concrete motor patterns is unknown.

The MB compartments are arranged such that the valence of DANs is opposite to that of the corresponding MBONs in a given compartment (*Aso et al., 2014b*). During learning, coactivation of DANs and KCs induces long-term depression of KC-MBON synapses in a compartment-specific manner (*Berry et al., 2018*; *Cohn et al., 2015*; *Hige et al., 2015a*; *Owald et al., 2015*). Thus, the prevailing hypothesis is that learning-induced depression in a subset of MBONs tips the collective balance of positive and negative valences represented by the MBON population, which are in balance in naive flies, and thereby biases the odor choice (*Heisenberg, 2003*; *Hige, 2018*; *Modi et al., 2020*; *Owald and Waddell, 2015*). Supporting this view, photoactivation of multiple types of MBONs encoding the same or opposite valences exerts additive effects for induction of attraction and avoidance (*Aso et al., 2014b*). This model predicts that the circuits downstream of the MB should be sensitive to skewed activity patterns of the MBON population. Such a computation can be performed by neurons integrating or comparing the output signals of multiple MBONs. In fact, axon terminals of the MBONs are confined to relatively limited brain regions, suggesting that they converge on common neurons (*Aso et al., 2014a*). The comprehensive EM connectome indeed revealed that 600 out of 1550 postsynaptic neurons of MBONs also receive input from at least one other MBON (*Li et al., 2020*). However, whether those convergent circuit motifs function to decode the parallel memories formed in the MB and, if so, how they shape motor patterns during memory retrieval are unknown.

The functional diversity of the MB compartments is not limited to the sign of memory valence. At least 5 out of 15 MB compartments are identified as appetitive memory compartments, and yet they exhibit distinct memory properties (*Aso et al., 2014b*; *Aso and Rubin, 2016*). For example, memory formation in the α1 compartment requires relatively long training, but once formed, lasts more than a day. In contrast, memory in γ5β′2a requires only a single training to form but is transient and easily overwritten by the subsequent training (*Aso and Rubin, 2016*; *Ichinose et al., 2021*; *Yamada et al., 2023*). Compartments are also tuned to distinct types of reward. While α1 memory is essential for nutritional value learning (*Yamagata et al., 2015*), γ4 and β′2 compartments function in water reward learning (*Lin et al., 2014*). Despite this diversity, memory formation in any appetitive compartments

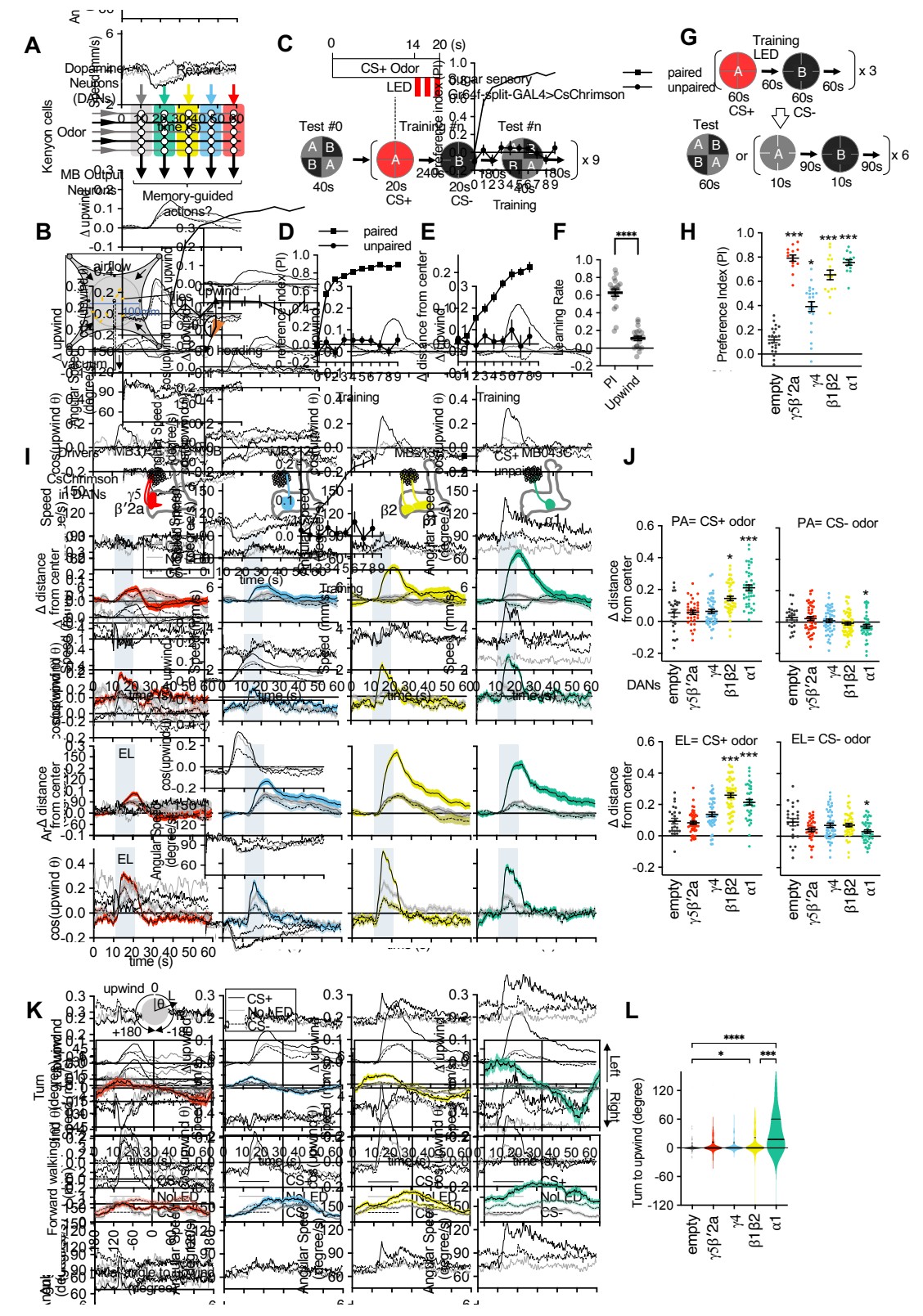

**Figure 1.** Memories in specific set of mushroom body (MB) compartments drive upwind locomotion. (**A**) A conceptual diagram of the MB circuit. The colored rectangles represent individual MB compartments. (**B**) A diagram of a four-armed olfactory arena. In each experiment, approximately 20 female flies were introduced into the circular arena. (**C**) Protocols for optogenetic training and tests used. One of two odors (pentyl acetate [PA] and ethyl lactate [EL]) was presented for 20 s, and three 1 s pulses of 627 nm light were started at 14 s. Another odor was presented alone, and then preference

*Figure 1 continued on next page*

*Figure 1 continued*

between two odors was measured. The cycles of training and tests were repeated nine times. In the unpaired protocol, LED was started 90 s before the onset of 'CS+' odor. (**D**) Preference to the CS+ odor in binary choice. (**E**) Displacement of flies' position relative to the center of the arena during the initial 14 s of 20 s odor period as wind-directional response. (**F**) Learning rate defined as response after single training divided by peak response after 9× training. (**G**) Protocols for optogenetic training and the two different memory tests used in this work. (**H**) Appetitive memories assessed by binary choice between CS+ and CS− odors immediately after training with optogenetic activation of dopaminergic neurons (DANs) that express CsChrimson with drivers indicated in (I). Empty is a split-GAL4 driver without promoters for AD and DBD domains. Thick and thin horizontal lines represent means and SEMs. Dunn's multiple comparison tests compared to empty-split-GAL4 control, following Kruskal-Wallis test; n=12–22. (**I**) Time course of the area-normalized mean of fly's position relative to the center of the arena as compared with its mean position at odor onset and the cosine of the angle between the fly's orientation and the upwind direction (see Materials and methods). Flies of each genotype were trained with three protocols: (1) pentyl acetate (PA) was paired with the LED illumination and ethyl lactate (EL) was unpaired. (2) EL was paired with LED and PA was unpaired. (3) Neither odor was paired (No LED). Lines and filled areas around lines are mean and SEM. N=24–60. (**J**) The delta of distance from the center at the end of the 10 s odor period. Each dot represents data from individual trials. Black lines are mean and SEM. *, p<0.05; ***, p<0.001; Dunn's multiple comparison tests compared to empty-split-GAL4 control, following Kruskal-Wallis test; n=24–60. The upwind displacements of MB213B and MB043C in response to CS+ odor were also significantly higher than the control when trial averages of six movies were compared. (**K**) Cumulative angle of turning and forward walking speed during the first 10 frames (333 ms; a time window we used for optogenetic experiments in *Figure 6*) following odor onset are plotted against initial angle to upwind, smoothened with ±30 degree bin. The number of trajectories analyzed for (CS+, CS−, No LED) conditions for MB109B+MB315C, MB312C, MB213B, and MB043C were (531, 562, 167), (710, 758, 814), (920, 1039, 919), and (449, 768, 531), respectively. Only flies that were 3 mm or more from the edge of the arena were analyzed. (**L**) The violin-plots of the cumulative angle of turn to the upwind orientation during the first 10 frames (333 ms) of odor onset. Only flies that oriented −90 to −150 or +90–150 degrees to the upwind direction at odor onset were analyzed. n=122, 137, 233, 239, 99 for empty-split-GAL4, MB109B+MB315C, MB312C, MB213B, and MB043C, respectively. *, p<0.05; ***, p<0.001; Dunn's multiple comparison tests compared to empty-split-GAL4 control, following Kruskal-Wallis test.

The online version of this article includes the following source data and figure supplement(s) for figure 1:

**Source data 1.** The values used for *Figure 1D*.

**Source data 2.** The values used for *Figure 1E*.

**Source data 3.** The values used for *Figure 1F*.

**Source data 4.** The values used for *Figure 1H*.

**Source data 5.** The values used for *Figure 1I*.

**Source data 6.** The values used for *Figure 1J*.

**Source data 7.** The values used for *Figure 1K*.

**Source data 8.** The values used for *Figure 1L*.

**Figure supplement 1.** Memory-based modulation of walking speed and angular speed depends on the fly's initial angle to the upwind direction.

**Figure supplement 1—source data 1.** The values used for *Figure 1—figure supplement 1*.

can promote attraction to the associated odor. However, the behavioral strategies used to find the source of attractive odors are not analyzed in typical olfactory learning assays using T-maze (*Tully and Quinn, 1985*), which only measure the relative distributions of flies between learned and control odors. Thus, the roles played by individual appetitive memory compartments in guiding approach to an attractive odor remain unknown.

By analyzing walking trajectories of individual flies, we found that appetitive memories formed in the α1 compartment are able to bias the turning direction so that flies move upwind. By photoactivation screening, we identified a single cluster of neurons, UpWind Neurons (UpWiNs), that can promote robust upwind steering and acceleration of locomotion. UpWiNs receive inputs from several types of lateral horn neurons and integrate inhibitory and excitatory inputs from MBON-α1 and MBON-α3, which are the output neurons of MB compartments that store long-lasting appetitive or aversive memories, respectively (*Aso and Rubin, 2016*; *Ichinose et al., 2015*; *Jacob and Waddell, 2022*; *Pai et al., 2013*; *Yamagata et al., 2015*). UpWiNs enhance responses to odors after induction of memory in the α1, and the activity of UpWiNs is required for appetitive memory and memory-driven upwind locomotion. Taken together, our work provides important insights into the process of valence integration, which we show employs a convergent circuit motif commonly found downstream of memory centers, and reveals circuit mechanisms that underlie the gradual transformation from abstract valence to specific motor commands.

## Results

### Identification of the MB compartments that drive upwind locomotion

To analyze behavioral components of memory-driven odor response, we used a modified four-armed olfactory arena in which odors are delivered through the current of airflow from the four channels at corners to the suction tubing at the center (*Figure 1B*; *Aso and Rubin, 2016*; *Pettersson, 1970*; *Vet et al., 1983*). The airstream from each channel forms sharp boundaries at the border of quadrants. Each of four quadrants can be filled with an arbitrary odor, but we typically used it either for presentation of a single odor in all quadrants or for binary choice by presenting two odors in diagonal quadrants (*Figure 1C*). Olfactory memories can be assessed by binary choice between two odors or by analyzing kinematic parameters and wind-directional behaviors in the presence of learned odor. When flies were repeatedly trained by pairing one of two odors with optogenetic activation of sugar sensory neurons, flies gradually increased upwind locomotion in response to the paired odor but developed odor preference in the binary choice more rapidly (*Figure 1D–E*). As a result, the learning rate measured by odor preference was much faster than that measured by upwind locomotion (*Figure 1F*). This observation and distinct dynamics of memory and plasticity in the MB lobes and compartments (*Aso et al., 2012*; *Aso and Rubin, 2016*; *Hige et al., 2015a*; *Huetteroth et al., 2015*; *Ichinose et al., 2021*; *Ichinose et al., 2015*; *Jacob and Waddell, 2022*; *Pai et al., 2013*; *Pascual and Préat, 2001*; *Plaçais et al., 2013*; *Séjourné et al., 2011*; *Vrontou et al., 2021*; *Yamagata et al., 2015*; *Zars et al., 2000*) led us to hypothesize that appetitive olfactory memories created in different MB compartments elicit distinct behaviors during memory retrieval.

To test the hypothesis, we first trained flies by pairing an odor as the CS+ with optogenetic activation of one of four sets of DANs. Each set of DANs projects to distinct appetitive memory compartments: γ5β′2a, γ4, β1β2, or α1 (*Huetteroth et al., 2015*; *Ichinose et al., 2015*; *Lin et al., 2014*; *Liu et al., 2012*; *Yamagata et al., 2015*). A second odor was presented without DAN activation as CS− (*Figure 1G*). These optogenetic activations promoted local release of dopamine in the targeted MB compartments (*Sun et al., 2020*; *Yamada et al., 2023*). We used a pair of odors, pentyl acetate (PA) and ethyl lactate (EL), that evokes activity in discrete sets of KCs (*Campbell et al., 2013*). After three training sessions, flies exhibited strong preference to the CS+ odor when given a choice between CS+ and CS− odors (*Figure 1H*). Next, we asked if these MB-compartment-specific memories can drive wind-directed movement when CS+ or CS− odors were presented separately for 10 s (*Figure 1G*). We measured the movement of individual flies and their heading angle relative to the upwind direction and analyzed how those parameters changed in response to odors. Despite robust CS+ preference in a binary choice, memories in the γ5β′2a and the γ4 failed to promote significant upwind movement (*Figure 1I–J*). In contrast, memories in the α1 and the β1β2 promoted steering and walking upwind in response to the CS+ odor, compared to genetic controls and the 'No LED' control group of the same genotype (*Figure 1I–J*). The memory in the α1 compartment also reduced upwind locomotion in response to CS− odor compared to the genetic control groups. These initial analyses compared averages of all ~20 flies in each movie. By separately analyzing behaviors of individual flies based on their orientation at the onset of odors, we found that memories in the α1 and β1β2 biased the direction of turning to steer toward upwind (*Figure 1K–L*). Memories in γ5β′2a and γ4 did not bias the turning direction, although they promoted flies initially facing downwind to change orientation in a non-directional manner (*Figure 1K–L* and *Figure 1—figure supplement 1*) and flies tended to orient toward upwind during CS+ odor presentation (*Figure 1I*). These results indicate that appetitive memory retrieval involves distinct behavioral strategies depending on the localization of the memory in the MB. Specifically, we expected that MBONs from the α1 and the β1β2 compartments are preferentially connected to circuit components that drive memory-driven upwind steering.

### Identification of UpWiNs by optogenetic screening

We next set out to identify the circuit elements that function downstream of the MBONs to induce memory-driven, wind-guided locomotion. To enable cell-type-specific experimental manipulation, we have made a large collection of split-GAL4 drivers (*Shuai et al., 2023*). Using a subset of these lines, we conducted optogenetic screening to test if activation of certain neurons can promote wind-directed movement. We analyzed how starved flies respond to 10 s optogenetic stimulation of various cell types in the circular arena with airflow but no olfactory stimuli (*Figure 2*). We measured the changes in the fly's distance from the center, heading angle relative to the upwind direction, angular

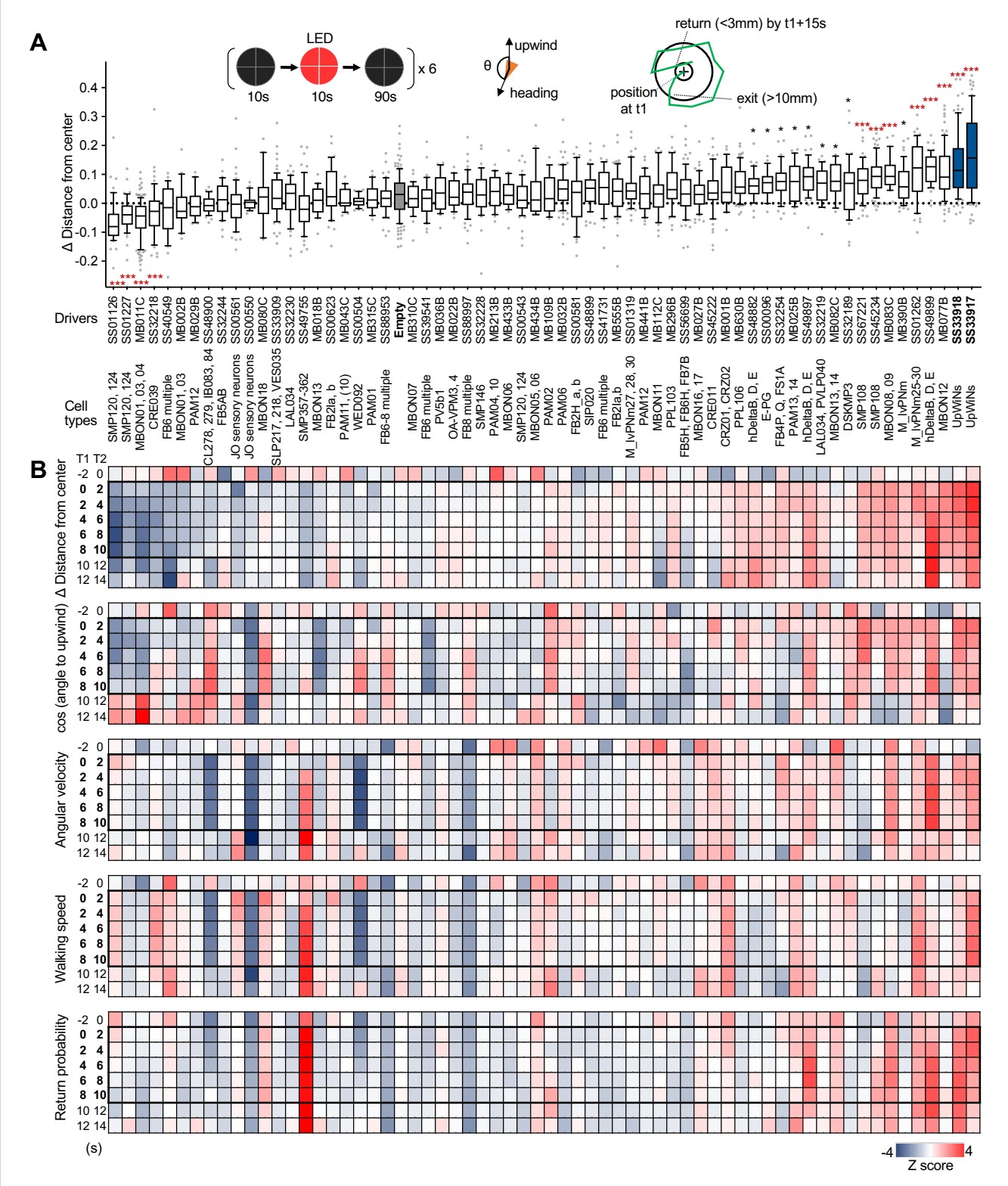

**Figure 2.** Identification of UpWind Neurons (UpWiNs) by activation screening. (**A**) Mean displacement of fly's position relative to the center of the arena during activation of various cell types defined by the indicated driver lines. Red asterisks indicate the results of Dunn's multiple comparison tests compared to empty-split-GAL4 control, following Kruskal-Wallis test;*, p<0.05; **, p<0.01; ***, p<0.001, n=18–132. Black asterisks indicate p<0.05 without correction for multiple comparisons. The median, first and third quartiles, 10 and 90 percentiles are displayed with outlier data points. Each of six movies from a group of flies was considered as a single data point. The conclusions about the UpWiNs lines (i.e. SS33917 and SS33918) did not

*Figure 2 continued on next page*

*Figure 2 continued*

change when trial averages of six movies were used for statistical tests. See *Figure 2—figure supplement 2* and http://www.janelia.org/split-gal4 for expression patterns of CsChrimson in these driver lines. (**B**) Z-scores for five parameters for 2 s time bins (T1 to T2) before, during (bold numbers), and after the 10 s activation period. Z-scores for driver line were calculated by (value – mean)/(standard deviation). For calculating the probability of return, 15-s-long trajectories of each fly following each time point (**t1**) were analyzed. A fly was considered to revisit the original location at time0 if it moved away more than 10 mm and came back to within 3 mm distance from that location at time0 within 15 s. 'time0' ranges 0–45 s, because the movies were 60 s long. High Z-score at 8–10 s time bin indicate that flies tended to move back to their location at 8–10 s by 23–25 s (i.e. mostly dark period after LED was turned off).

The online version of this article includes the following source data and figure supplement(s) for figure 2:

**Source data 1.** The values used for *Figure 2A*.

**Source data 2.** The values used for *Figure 2B*.

**Figure supplement 1.** Activation phenotypes of 'hit' lines.

**Figure supplement 1—source data 1.** The values used for *Figure 2–figure supplement 1*.

**Figure supplement 2.** Expression patterns of 'hit' lines.

**Figure supplement 3.** LM-EM matching of cell types in SS49899.

**Figure supplement 4.** LM-EM matching of cell types in SS49755.

velocity, and walking speed. Because returning to the odor plume is a major component of olfactory navigation (*Baker, 1990*; *Cardé, 2021*), we also measured the probability of a fly returning to its starting location after moving away.

Although our screening was not comprehensive in terms of the coverage of the cell types or brain areas, it successfully identified several clear 'hits', which include both known and previously uncharacterized cell types. Four lines, which label SMP120/124, MBON01/03/04, or CRE039, promoted locomotion in the downwind direction (*Figure 2* and *Figure 2—figure supplements 1 and 2*). As previously reported (*Matheson et al., 2022*), activation of some MBON types including MBON-α3 (also known as MBON14 or MBON-V3) and MBON-γ2α'1 (MBON12) promoted significant upwind locomotion. *Figure 2B* summarizes the detailed time courses of these behavioral phenotypes before, during, and after 10 s LED stimulations. These behavioral data can be immediately put into the context of the EM connectome map, since the cell types in each driver lines were morphologically matched by comparing confocal and electron microscope images (see examples in *Figure 2— figure supplements 3 and 4*).

Among the split-GAL4 drivers we screened, SS33917 and SS33918 showed the strongest upwind locomotion, especially at the onset of 10 s activation period (*Videos 1 and 2*; *Figure 2B* and *Figure 2—figure supplement 1*). These driver lines label a similar set of 8–11 neurons (*Figure 2—figure supplement 2*). Here, we will focus our analysis on this cluster of neurons, which we collectively call UpWiNs, based on their robust activation phenotype and anatomical connections with the α1 compartment (see below).

## UpWiNs integrate inputs from MBONs

The UpWiNs have extensive arborizations in the posterior dorsolateral area of the brain where MBON-α1 (also known as MBON07) and MBON-α3 (also known as MBON14) send converging axons (*Video 3*; *Figure 3A*). DANs innervating the α1 and α3 compartments respond to sugar or electric shock/heat/bitter, and activation of DANs can

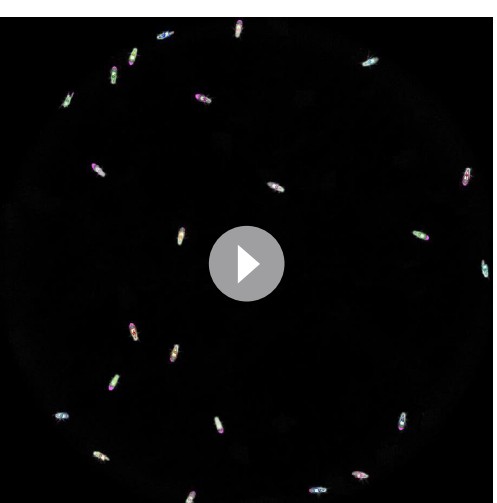

**Video 1.** Activation phenotype of UpWind Neurons (UpWiNs). An example movie of UpWiN activation in SS33917>CsChrimson flies used in *Figure 2*. The red square at the bottom right corner indicates the 10 s period when the red LED was turned on. The small circles indicate the centroid of flies and triangles indicate the orientation of flies. The diameter of the arena is 10 cm.

https://elifesciences.org/articles/85756/figures#video1

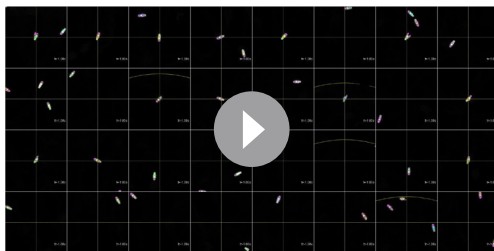

**Video 2.** Activation phenotype of UpWind Neurons (UpWiNs) depends on the initial orientation. Cropped movies of individual SS33917>CsChrimson flies centered and reoriented based on the position and the angle to upwind at the onset of the activating illumination (related to *Figure 2* and *Figure 6D–G*). The red square at the bottom right corner of each panel indicates when the red LED was turned on. The small circles indicate the centroid of flies and triangles indicate the orientation of flies. The airflow direction was from the top to the bottom of each panel (2x2cm). https://elifesciences.org/articles/85756/figures#video2

substitute US to induce long-lasting appetitive and aversive memories, respectively (*Aso and Rubin, 2016*; *Huetteroth et al., 2015*; *Ichinose et al., 2015*; *Jacob and Waddell, 2022*; *Kirkhart and Scott, 2015*; *Matheson et al., 2022*; *Pai et al., 2013*; *Schnitzer et al., 2022*; *Siju et al., 2020*; *Vrontou et al., 2021*; *Yamagata et al., 2015*). MBON-α1 displays reduced odor response to an odor associated with activation of DANs in α1 (*Yamada et al., 2023*), whereas MBON-α3 increases response to an odor associated with sugar reward possibly due to interactions with appetitive memory compartments such as the β1 (*Li et al., 2020*; *Plaçais et al., 2013*; *Takemura et al., 2017*; *Tanaka et al., 2008*) and decreases response to punishment-associated odors (*Jacob and Waddell, 2022*; *Schnitzer et al., 2022*). Both MBONs are required for retrieval of long-term appetitive memory (*Ichinose et al., 2015*; *Plaçais et al., 2013*). These previous reports raise the possibility that the UpWiNs defined by the SS33917 driver might play a role in both the upwind locomotion observed during retrieval of an α1 memory (*Figure 1*) and the activation of MBON-α3 (*Matheson et al., 2022*) (MB082C data in *Figure 2*).

To test this possibility, we first examined the anatomical connectivity of the UpWiNs. We obtained images of 25 individual neurons in SS33917-split-GAL4 by the multi-color flip-out method and compared them with reconstructed EM-images (*Figure 3* and *Figure 3—figure supplements 1–3*; *Nern et al., 2015*; *Otsuna et al., 2018*; *Scheffer et al., 2020*). This analysis identified 11 neurons of five cell types in the hemibrain EM dataset that resemble UpWiNs in SS33197 driver (*Figure 3—figure supplements 1 and 2*). Among 11 matched EM-reconstructed neurons of the UpWiNs, four neurons, one SMP353 and three SMP354 neurons, receive direct synaptic input from MBON-α1 (*Figure 3B–E*; *Li et al., 2020*; *Scheffer et al., 2020*). SMP354 also receives input from MBON-α3; this strong convergent connectivity is exceptional among the population of the neurons that are postsynaptic to either of the MBONs (*Figure 3D–E* and *Figure 3—figure supplement 4*). The rest of the UpWiNs do not have direct connections with these MBONs but receive indirect input from them via connections among UpWiNs (*Figure 3F*). The interconnection within the UpWiN cluster suggests that these neurons may function as a group, even though the connectivity of the individual neurons is heterogeneous. Interestingly, all the UpWiNs provide input to a single neuron, SMP108 (*Figure 3C*), which has the highest number of connections with reward DANs and plays a key role in second-order conditioning (*Yamada et al., 2023*). The axon terminals of UpWiNs are immunoreactive to choline acetyltransferase (*Yamada et al., 2023*), and therefore likely to be excitatory to the SMP108 and other downstream neurons. The SMP108 is labeled in SS45234 and SS67221, and its activation also promoted upwind locomotion (*Figure 2*; *Figure 2—figure supplement 1*).

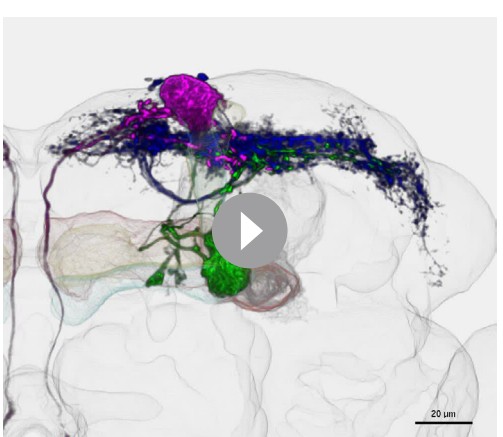

**Video 3.** Convergent projection of mushroom body output neuron (MBON)-α1 and MBON-α3 onto the dendritic area of UpWind Neurons (UpWiNs). Overlay of MBON-α1, MBON-α3, and UpWiNs in a standard brain. UpWiNs were originally identified by searching neurons that overlap with convergent axonal projection of these MBONs using a database of confocal microscope images to generate split-GAL4 driver lines. https://elifesciences.org/articles/85756/figures#video3

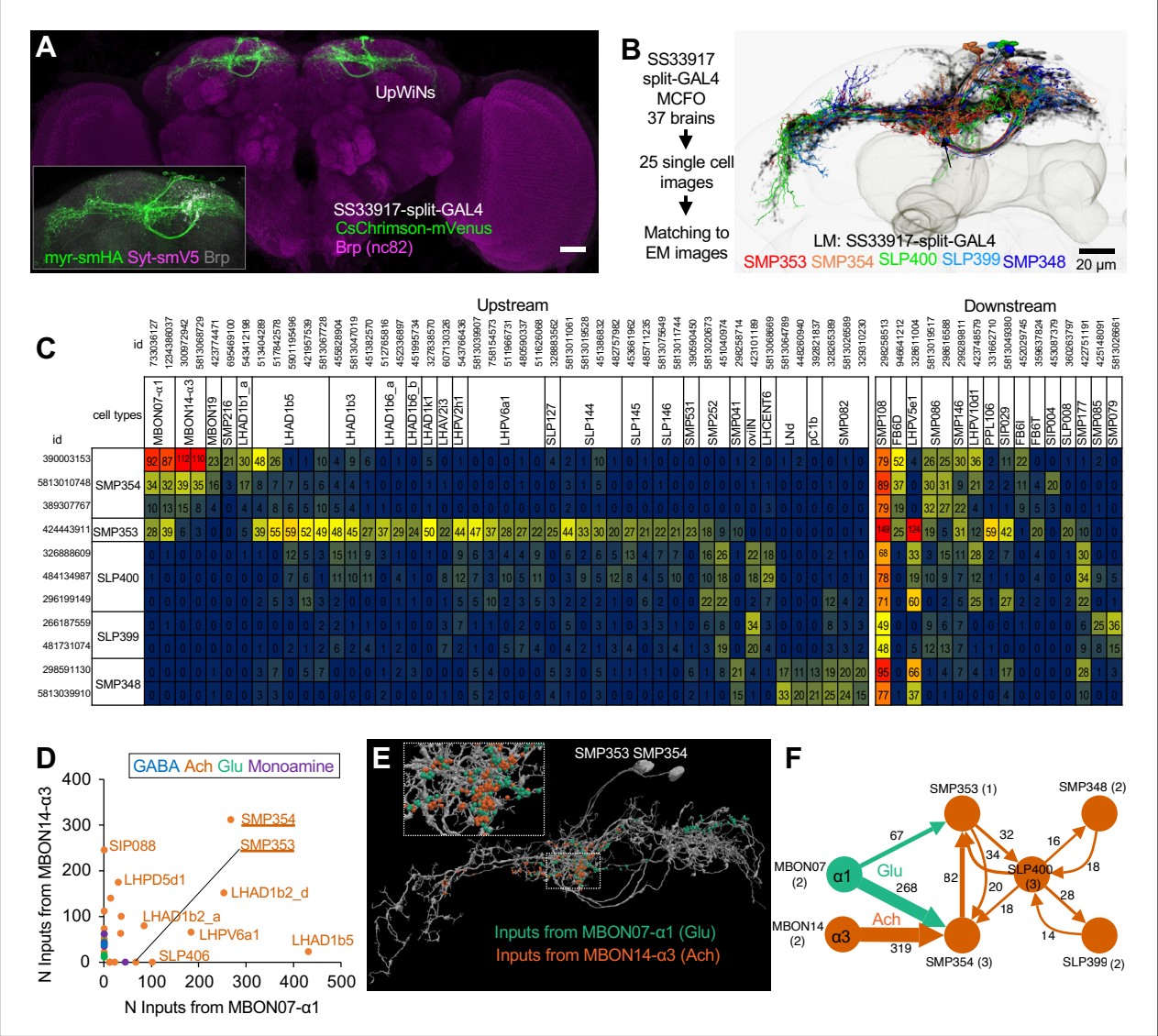

**Figure 3.** Connectivity of UpWind Neurons (UpWiNs). (**A**) The expression pattern of CsChrimson-mVenus driven by split-GAL4 line SS33917. The scale bar is 20um. The insert image shows signals of membrane reporter myr-smHA and presynaptic reporter Syt-smV5 driven by the same driver. (**B**) Eleven EM-reconstructed neurons that correspond to UpWiNs defined by the SS33917 driver were identified by analyzing the morphology of individual neurons (*Figure 3—figure supplements 1 and 2*) and are displayed with outline of the MB and the standard brain. Individual neurons are color-coded to indicate the cell type to which they were assigned. (**C**) Connectivity of UpWiNs with major upstream and downstream neurons that have at least 20 connections with 1 of the 11 UpWiNs. The hemibrain body IDs of each neuron is shown as well as their assignment to specific cell types. Numbers indicate the number of synapses from the upstream neurons to UpWin neurons (left) or from the UpWiNs to the downstream neurons (right). (**D**) Interneurons downstream to mushroom body output neuron (MBON)-α1 and MBON-α3. Colors of dots indicate neurotransmitter prediction (*Eckstein et al., 2020*). See *Figure 3—figure supplement 3* for more details. (**E**) Predicted postsynaptic sites in SMP353 and SMP354 (gray), which are juxtaposed to presynaptic sites from MBON-α1 (green) and MBON-α3 (orange). The insert (10 μm width) shows a magnified view of juxtaposed synapses. (**F**) Interconnectivity between UpWiNs. The numbers indicate the summed number of connections. The numbers in parentheses indicate the number of neurons per cell type.

The online version of this article includes the following figure supplement(s) for figure 3:

**Figure supplement 1.** Candidate UpWind Neurons (UpWiNs) in hemibrain EM images.

**Figure supplement 2.** Single cell images of neurons in SS33917.

**Figure supplement 3.** NBLAST clustering of single cell images of neurons in SS33917.

**Figure supplement 4.** Downstream neurons of mushroom body output neuron (MBON)-α1 and MBON-α3.

To test functional connectivity, we made in vivo whole-cell recordings from UpWiNs while optogenetically activating either MBON-α3 or MBON-α1. Neurons were randomly targeted by the electrode among those labeled by R64A11-LexA, which is a broad driver for UpWiNs. R64A11 is a hemi-driver for the DNA-binding domain of the SS33917-split-GAL4. A brief 10 ms stimulation of cholinergic MBON-α3 evoked a strong excitation in 3 out of 11 UpWiNs examined, whereas glutamatergic MBON-α1 evoked inhibitory responses in 4 out of 17 UpWiNs (*Figure 4A–B*). The observed stochasticity of the connectivity is consistent with the EM connectome data.

Postsynaptic sites of MBON-α1 and MBON-α3 are juxtaposed on the dendrites of UpWiNs (*Figure 3E*), implying dendritic integration of these inputs. Since we did not have a LexA driver that selectively labels SMP354, we were unable to specifically target those integrating UpWiNs by electrophysiology. We therefore measured the population activity of UpWiNs at the junction between their dendrites and proximal axons by two-photon calcium imaging in dissected brains. Consistent with the electrophysiological results and the circuit model, we observed a calcium increase upon MBON-α3 activation. Moreover, MBON-α1 activation suppressed the excitatory effect of MBON-α3 when they were activated together (*Figure 4C*).

Finally, we tested the presence of excitatory interconnection between UpWiNs. We expressed GCaMP6s in a broad population of UpWiNs using 64A11-LexA while expressing Chrimson-tdTomato in a small subset using SS67249 split-GAL4. The flies also carried UAS-LexAp65-DBD2-RNAi to suppress the expression of GCaMP in Chrimson-positive UpWiNs (*Figure 4—figure supplement 1A–B*). 1 s photostimulation evoked excitatory GCaMP response in both axons and dendrites (*Figure 4—figure supplement 1C*). These results indicate that as a population, UpWiNs receive and integrate synaptic inputs from MBONs that signal opposite signs of memory valence.

## UpWiNs acquire enhanced responses to reward-predicting odors

The UpWiN cluster collectively receives olfactory information from the MBONs and lateral horn output neurons (*Figure 3C*). This anatomy raises the intriguing possibility that UpWiNs have basal odor responses and that memories in the MB modify it. To test this possibility, we optogenetically induced appetitive memory and monitored the change in the subsequent odor-evoked electrophysiological activity of UpWiNs (*Figure 5A*). For these experiments, we used another UpWiNs split-GAL4 driver SS67249. This driver was not suitable for behavioral experiments due to stochastic and off-targeted expression but labeled a highly restricted subset (one to three cells) of UpWiNs including the one resembling the morphology of SMP353 (*Figure 5—figure supplement 1*). Before training, the UpWiNs showed relatively weak odor responses (*Figure 5B*) likely because inhibitory and excitatory inputs cancel each other (*Figure 4C*). After pairing an odor with optogenetic activation of reward DANs including those projecting to α1, UpWiNs displayed increased excitatory response to subsequent exposures to the CS+ odor but not to the CS− odor (*Figure 5B–D*). We observed the enhancement of CS+ response irrespective of the identity of tested CS+ odors (OCT or MCH; *Figure 5—figure supplement 2*). This enhancement of CS+ response can be most easily explained as an outcome of disinhibition from MBON-α1 whose output had been decreased by memory formation; MBON-α1 is inhibitory to UpWiNs (*Figure 4B*) and MBON-α1 response to the CS+ is reduced following the same training protocol (*Yamada et al., 2023*). In addition to such a mechanism, plasticity in the β1 compartment may also contribute to the enhanced CS+ response in UpWiNs because the driver R58E02-LexA contains DANs in the β1 and glutamatergic MBON from the β1 directly synapses on the dendrites of MBON-α1 and MBON-α3 (*Takemura et al., 2017*).

## UpWiNs promote wind-directed behaviors

Having examined the functional connectivity and plasticity of UpWiNs, we revisited behavioral phenotypes caused by optogenetic activation. In the screening experiments shown in *Figure 2*, since the default wind direction is from the periphery to the center in our olfactory arena, upon activation of UpWiNs, flies moved toward the periphery and increased their mean distance from the center. However, this phenotype might be explained by the avoidance of center area (*Besson and Martin, 2005*) rather than wind-directed behavior. Several experiments argue against that possibility. First, the UpWiN activation phenotype was starvation dependent; only starved flies showed robust upwind locomotion upon UpWiN activation (*Figure 6A*). Second, flies' response to UpWiN activation depended on the rate and direction of the airflow. Flies did not move toward the periphery without airflow (*Video 4*) and

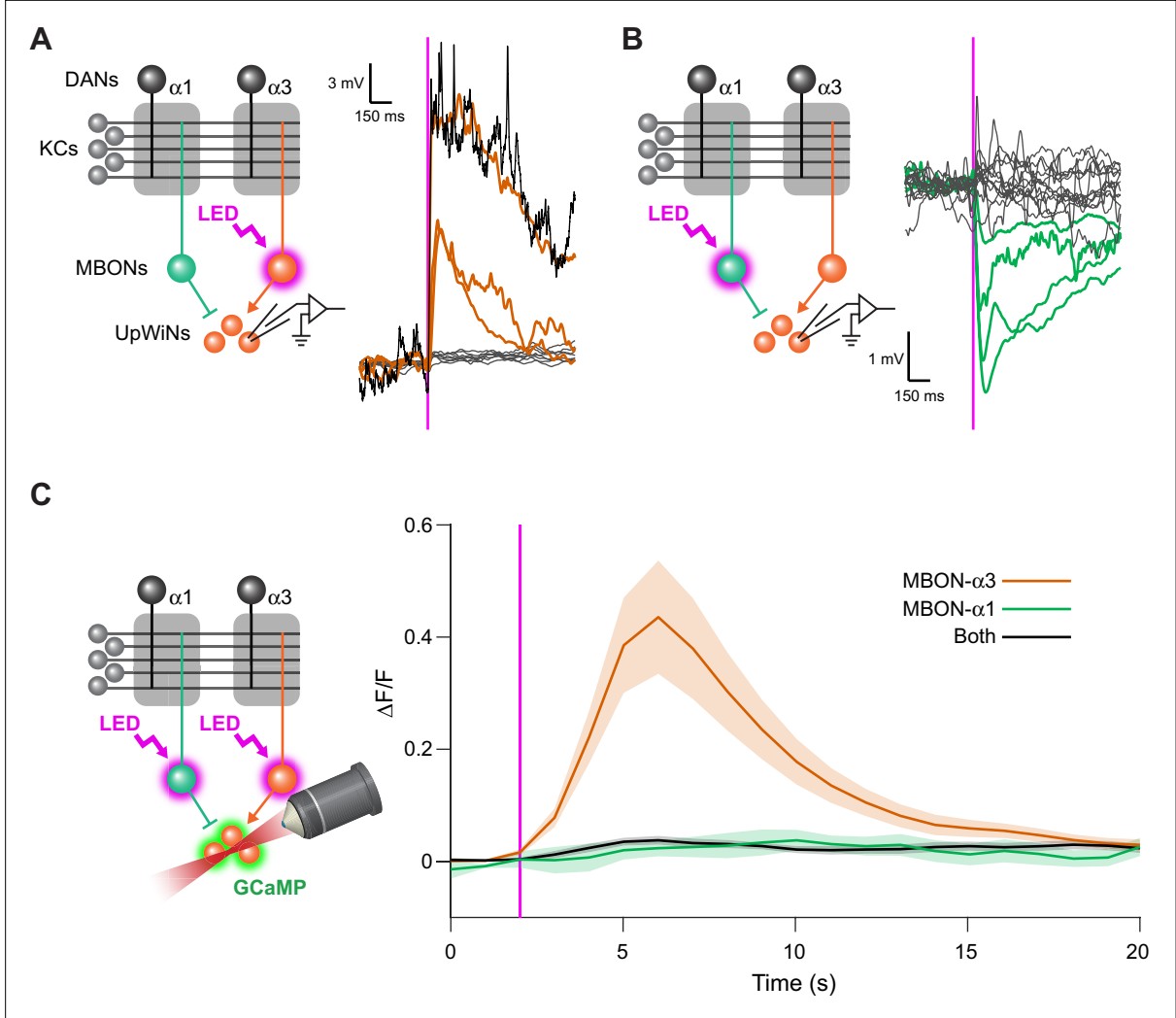

**Figure 4.** UpWind Neurons (UpWiNs) integrate excitatory and inhibitory synaptic inputs from mushroom body output neurons (MBONs). (**A**) Functional connectivity between MBON-α3 and UpWiNs. Chrimson88-tdTomato was expressed in MBON-α3 by MB082C split-GAL4, and the photostimulation responses were measured by whole-cell current-clamp recording in randomly selected UpWiNs labeled by R64A11-LexA. 3 out of 11 neurons (7 flies) showed excitatory response. Mean voltage traces from individual connected (orange) and unconnected UpWiNs (gray) are overlaid. The connection was strong enough to elicit spikes (black; single-trial response in one of the connected UpWiNs). Magenta vertical line indicates photostimulation (10 ms). (**B**) Functional connectivity between MBON-α1 and UpWiNs. Chrimson88-tdTomato expression in MBON-α1 was driven by MB310C split-GAL4. 4 out of 17 neurons (12 flies) showed inhibitory response. Mean voltage traces from individual connected (green) and unconnected UpWiNs (gray) are overlaid. (**C**) Integration of synaptic inputs from MBON-α3 and MBON-α1. Population responses of UpWiNs were measured by two-photon calcium imaging at the junction between dendrites and axonal tracts (mean ΔF/F ± SEM) while photostimulating MBON-α3 (orange; n=5), MBON-α1 (green; n=11) or both (black; n=7). Expression of GCaMP6s was driven by R64A11-LexA, and Chrimson88-tdTomato to G0239-GAL4 (MBON-α3) and/or MB310C (MBON-α1). Photostimulation: 1 s (magenta). While activation of MBON-α1 did not evoke detectable inhibition in the calcium signal, it effectively canceled the excitation by MBON-α3.

The online version of this article includes the following source data and figure supplement(s) for figure 4:

**Source data 1.** The values used for *Figure 4A*.

**Source data 2.** The values used for *Figure 4B*.

**Source data 3.** The values used for *Figure 4C*.

**Figure supplement 1.** Excitatory interconnections between UpWind Neurons (UpWiNs).

**Figure supplement 1—source data 1.** The values used for *Figure 4—figure supplement 1*.

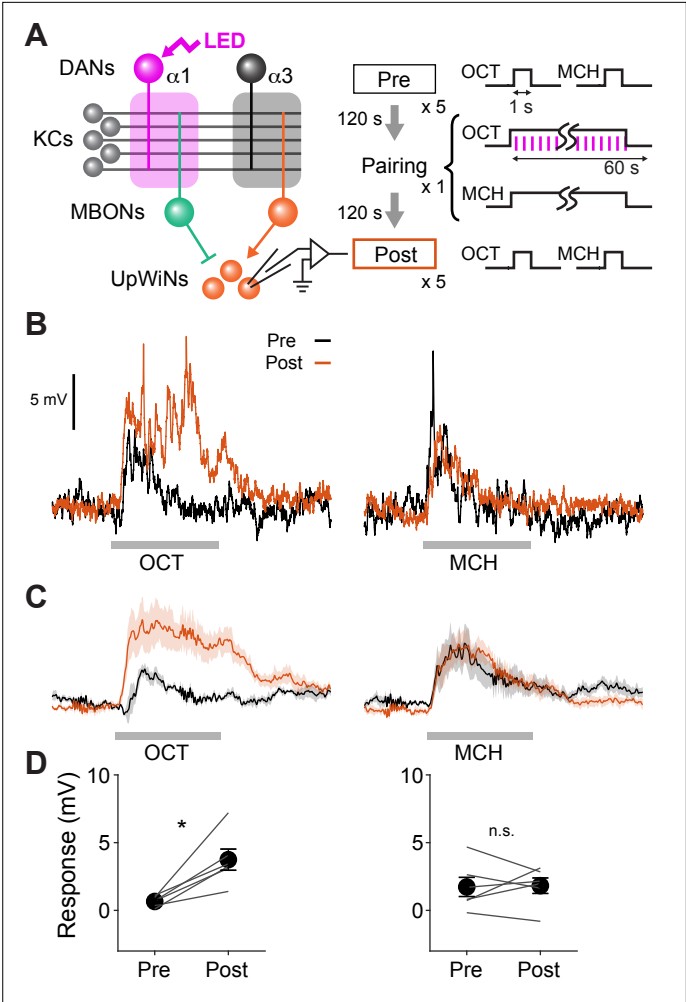

**Figure 5.** Optogenetic appetitive conditioning enhances the response to the conditioned odor in UpWind Neurons (UpWiNs). (**A**) Optogenetic conditioning was performed by pairing photostimulation of PAM-dopaminergic neurons (DANs) with odor presentation. Expression of ChrimsonR-mVenus was driven by 58E02-LexA, and in vivo whole-cell recordings were made from UpWINs labeled by GFP using SS67249-split-GAL4. 1 min presentation of OCT was paired with LED stimulation (1 ms, 2 Hz, 120 times), followed by 1 min presentation of MCH alone. (**B**) Representative recording from a single fly. Gray bars indicate 1 s odor presentation. (**C**) Mean (± SEM) odor responses (n=6). Spikes were removed by a low-pass filter. (**D**) Summary data of mean (± SEM) odor-evoked membrane depolarization. Gray lines indicate data from individual neurons. Responses to OCT were potentiated (p<0.01; repeated-measures two-way ANOVA followed by Tukey's post hoc multiple comparisons test), while those to MCH did not change (p=0.9).

The online version of this article includes the following source data and figure supplement(s) for figure 5:

**Source data 1.** The values used for *Figure 5B*.

**Source data 2.** The values used for *Figure 5C*.

**Source data 3.** The values used for *Figure 5D*.

**Figure supplement 1.** Expression patterns of SS67249.

**Figure supplement 2.** Reciprocal experiment of optogenetic appetitive conditioning.

**Figure supplement 2—source data 1.** The values used for *Figure 5—figure supplement 2B*.

**Figure supplement 2—source data 2.** The values used for *Figure 5—figure supplement 2C*.

**Figure supplement 2—source data 3.** The values used for *Figure 5—figure supplement 2C*.

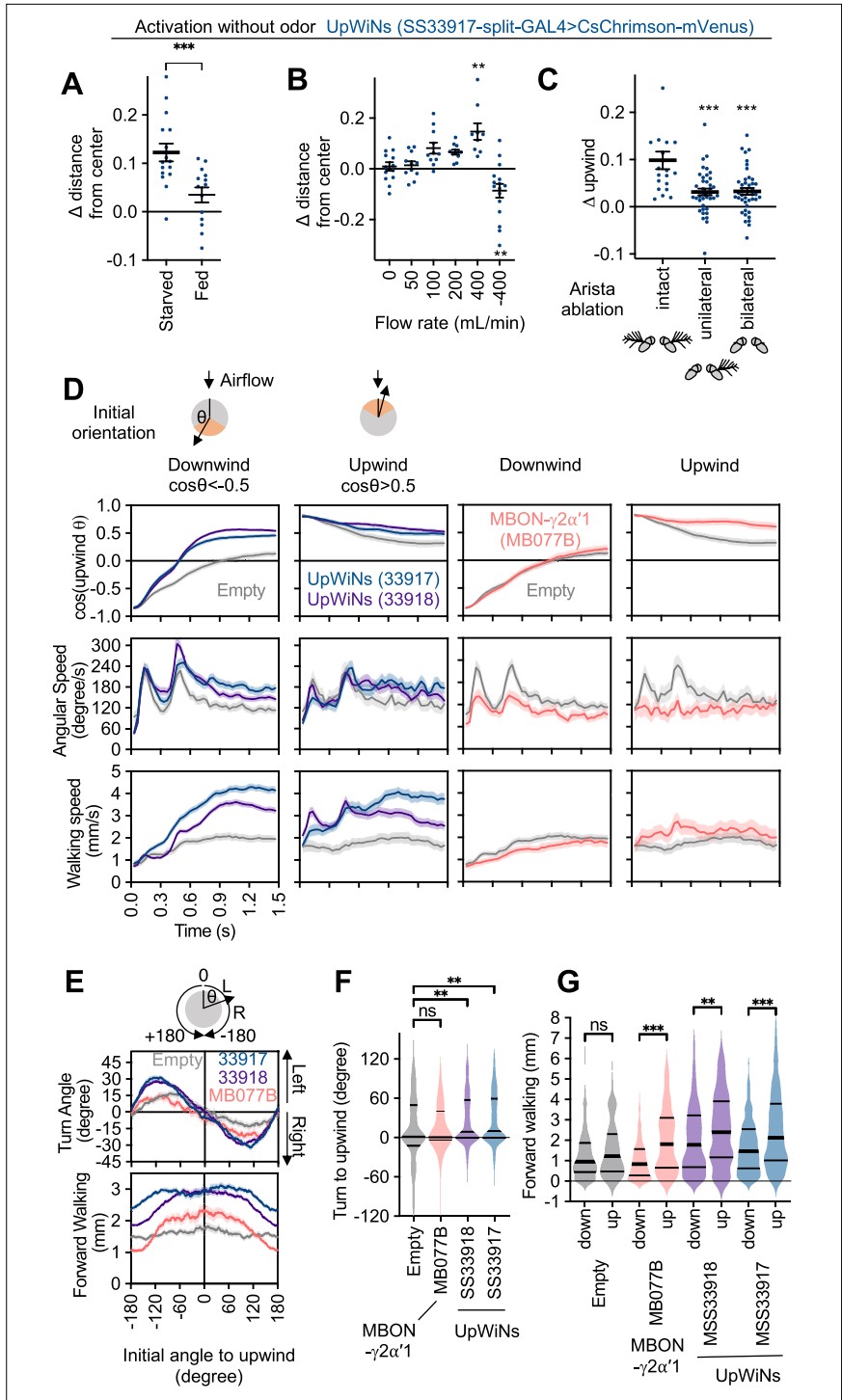

**Figure 6.** Activity of UpWind Neurons (UpWiNs) bias turning direction. (**A**) Fed or 40–48 hr starved flies were compared to assess requirement of starved status for UpWiNs to promote upwind locomotion. n=14 (fed) and 16 (starved); ***, p<0.001, Mann-Whitney test. (**B**) Upwind locomotion during the 10 s activation of UpWiNs in the arena with various rates of airflow. n=9–16;**, p<0.01; Dunn's multiple comparison tests compared to the zero flow condition. (**C**) Right side or both sides of aristae were ablated 1 day prior to experiments to measure upwind response during UpWiN activation. n=20 (intact) and 40 (unilateral and bilateral); ***, p<0.001; Dunn's multiple comparison tests compared to the intact control. (**D**) Behavioral kinematics of UpWiN activation. The trajectories of individual flies during first 1.5 s of 10 s LED period were grouped to initially facing downwind or upwind if cos(upwind angle) was above 0.5 or below –0.5, respectively. (**E**) Cumulative angle of turning and forward

*Figure 6 continued on next page*

*Figure 6 continued*

walking speed during the first 10 frames (333 ms) after the onset of LED plotted against initial angle to upwind smoothened with ±30 degree bin. The number of trajectories analyzed for (SS33917, SS33918, MB077B, empty-split-GAL4) were (2492, 3362, 772, 1582), respectively. Only flies that were at least 3 mm away from the edge of the arena were analyzed. (**F–G**) The violin-plots of the cumulative angle of turn to the upwind orientation or forward walking speed during the first 10 frames (333 ms) of odor onset. Only flies that oriented –90 to –150 or +90–150 degrees to upwind at the odor onset were analyzed. n=444, 540, 231, 219 for SS33917, SS33918, MB077B, empty-split-GAL4, respectively. **, p<0.01; ***, p<0.001; Dunn's multiple comparison for the selected pairs, following Kruskal-Wallis test. Thick and thin horizontal lines are mean and SEM in (**A–C**) and median and quartile ranges in (**F–G**), respectively.

The online version of this article includes the following source data and figure supplement(s) for figure 6:

**Source data 1.** The values used for *Figure 6A*.

**Source data 2.** The values used for *Figure 6B*.

**Source data 3.** The values used for *Figure 6C*.

**Source data 4.** The values used for *Figure 6D*.

**Source data 5.** The values used for *Figure 6E*.

**Source data 6.** The values used for *Figure 6F*.

**Source data 7.** The values used for *Figure 6G*.

**Figure supplement 1.** The cosine of angle to upwind, angular speed and forward walking speed are separately plotted for flies oriented downwind or upwind at the odor onset.

**Figure supplement 1—source data 1.** The values used for *Figure 6—figure supplement 1*.

moved toward the center when the direction of airflow was reversed (*Figure 6B*). Finally, unilateral, or bilateral ablation of aristae, the wind-sensing organ in *Drosophila* (*Yorozu et al., 2009*), impaired movement toward the periphery during UpWiN activation (*Figure 6C*). With unilateral ablation of arista, activation of UpWiNs still initiated turning but flies turned too much and failed to make a transition to forward walking toward upwind orientation (*Video 5*). These observations are consistent with a role for UpWiNs in transforming appetitive memory into wind-directed behaviors.

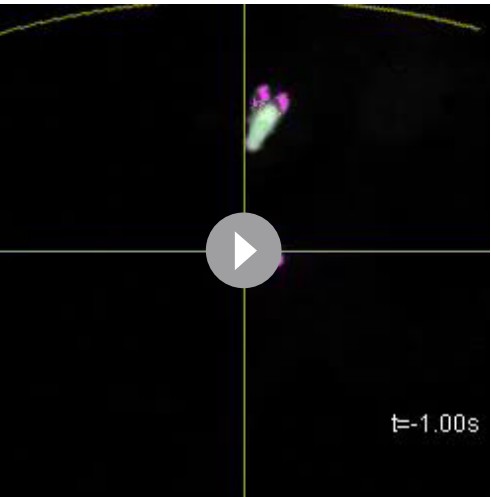

**Video 5.** Activation phenotype of UpWind Neurons (UpWiNs) depends on the intact aristae. An example UpWiN activation phenotype in an SS33917>CsChrimson fly that lacked the arista on the right side (related to *Figure 6C*). 2x2cm are was cropped and reoriented based on the position and the angle to upwind at the onset of the activating illumination. The red square at the bottom right corner indicates when the red LED was turned on. The small circles indicate the centroid of flies and triangles indicate the orientation of flies. The airflow direction was from the top to the bottom.

https://elifesciences.org/articles/85756/figures#video5

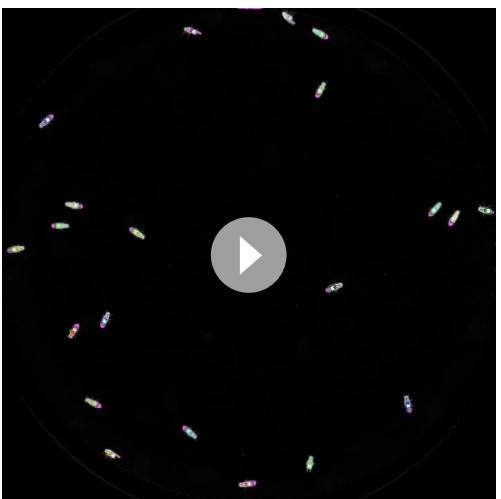

**Video 4.** Activation phenotype of UpWind Neurons (UpWiNs) without airflow. An example movie of UpWiN activation in SS33917>CsChrimson flies without airflow (related to *Figure 6B*).

https://elifesciences.org/articles/85756/figures#video4

As observed in memory-driven olfactory responses (*Figure 1* and *Figure 1—figure supplement 1*), the kinematics of behavior at the onset of UpWiN activation depended on the initial orientation of flies relative to the wind direction (*Video 2*). Flies transiently increased angular speed during the first ~300 ms (*Figure 6D*). This increased angular speed was observed also in empty-split-GAL4 control flies and considered to be a startle response to activating light. However, direction of turning during this period was significantly biased toward the upwind direction when either of two lines for UpWiNs were used to express CsChrimson (*Figure 6E–F*). UpWiNs activation also modulated forward walking speed in a manner that depended on the orientation of flies at the onset of the activating light (*Figure 6G*). The orientation-dependent modulation of turning direction and walking speed observed is similar to that evoked by α1-specific memory (*Figure 1K*). In contrast, activation of MBON-γ2α′1 with MB077B split-GAL4 modulated forward walking speed and promoted flies that already faced upwind to maintain that orientation but did not cause directional turning toward the upwind direction (*Figure 6D–G*; see *Figure 6—figure supplement 1* for other drivers). These results are consistent with a view that UpWiNs transform memory in an α1 into signals that promote olfactory navigation but do not yet specify lower-level motor parameters (i.e. turning direction and acceleration). Information about wind direction and UpWiN's activity needs to be integrated somewhere downstream to compute the turning direction. The central complex is the likely brain area for such a computation (*Matheson et al., 2022*; *Okubo et al., 2020*).

## UpWiNs are required for memory-driven upwind locomotion

Finally, we asked if UpWiNs are required for retrieval of sugar-induced appetitive memory. Formation of long-lasting appetitive memory after odor-sugar conditioning relies on the DANs that innervate the α1 compartment (*Ichinose et al., 2021*; *Yamagata et al., 2015*). Therefore, we tested the requirement of UpWiNs for 1 day appetitive memory. The control genotypes showed enhanced upwind locomotion in the presence of odors associated with sugar, whereas flies that express the light chain of tetanus toxin (TNT) in UpWiNs showed compromised upwind locomotion (*Figure 7A–B*). To test the requirement of UpWiNs specifically during the memory test period, we also attempted experiments with temperature-sensitive *shibire*, which allows reversible block of vesicular release (*Kitamoto, 2001*). One day after odor-sugar conditioning, blocking synaptic output of UpWiNs only during test period impaired preference to CS+ odor in binary choice compared to the genetic controls (*Figure 7C*). However, we were unable to analyze wind-directional behaviors in these *shibire* experiments because control flies did not show CS+ odor-induced upwind locomotion at restrictive temperature (data not shown) presumably due to increased preference to the peripheral of the arena or altered odor concentration. These results indicate that UpWiNs play a major role in behavior during appetitive memory retrieval but also suggest that their behavioral contribution may not be limited to simple promotion of upwind locomotion. Indeed, the analysis of 10 s activation screening data revealed that flies increased the probability of revisiting the location where UpWiNs activation was ended (*Figure 2B*). Although this data does not necessarily indicate induction of spatial memory by UpWiNs, revisiting behavior cannot be explained by a simple increase in turning probability. The return probability plotted in *Figure 7E* is probability of return to the position at the end of LED period within 15 s post-LED period when angular speed of SS33917>CsChrimson and SS33918>CsChrimson flies are identical to empty-split-GAL4>CsChrimson control flies (*Figure 7—figure supplement 1*). Another set of cell types SMP357-362 defined by SS49755-split-GAL4 caused far more robust revisiting phenotype (*Video 6*; *Figure 2B* and *Figure 2—figure supplement 2*). Finally, we found that optogenetic activation of the UpWiNs could bias spatial distribution of flies between quadrants with and without activating illumination (*Figure 7D*). This bias is likely due to the airflow-independent function of UpWiNs because the UpWiN activation could increase the probability of revisiting behavior even in the absence of airflow (*Figure 7E* and *Figure 7—figure supplement 1*).

## Discussion

It has been postulated that the valence of learned odors is represented as the relative activity of the MBONs, each of which signals either positive or negative valence assigned in the parallel memory modules. In this study, we identified a cluster of neurons that can decode the differential activity of MBONs encoding opposing valances. Although activity of these neurons strongly induced a

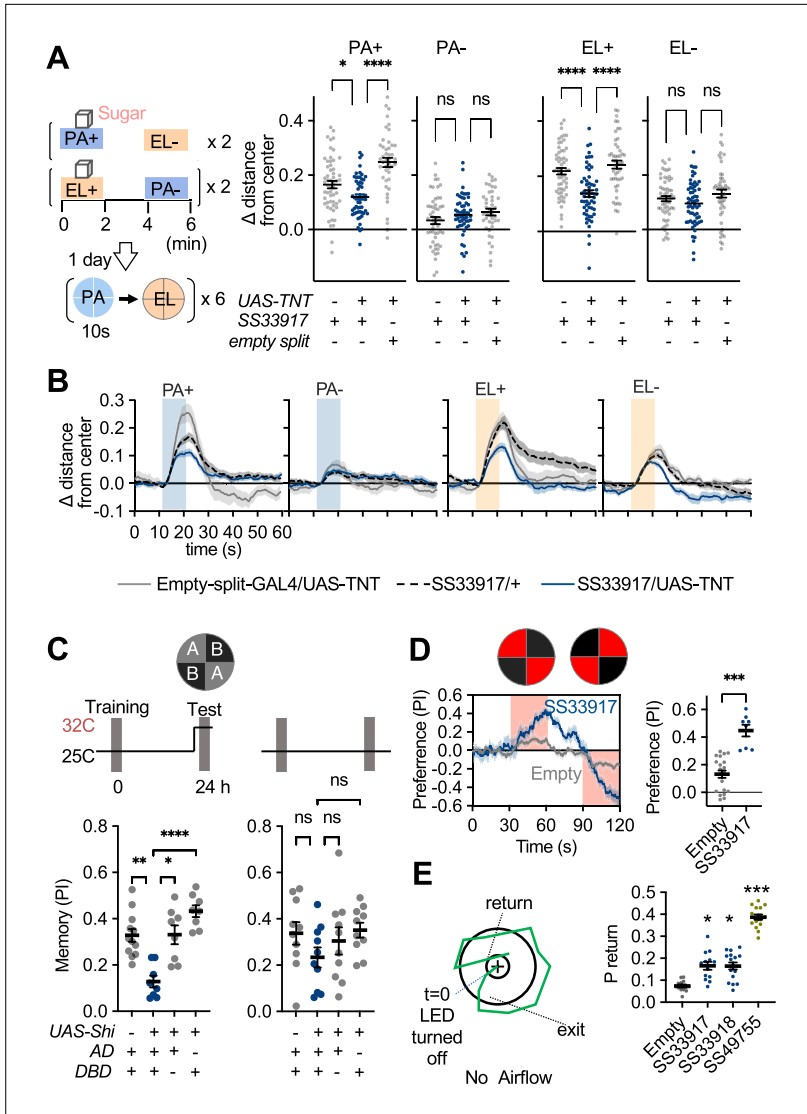

**Figure 7.** UpWind Neurons (UpWiNs) are required for memory-driven upwind locomotion. (**A**) Upwind response to the odor associated with sugar in control genotypes and flies that express tetanus toxin (TNT) in UpWiNs. (**B**) Time course of upwind response. (**C**) Appetitive memories of control genotypes and flies expressing shibire (*Shi*) in UpWiNs were tested 1 day after odor-sugar conditioning at restrictive or permissive temperature. (**D**) Time course of fly's preference to quadrants with red LED light by SS33917>CsChrimson (blue) or empty-split-GAL4>CsChrimson (gray). The preference to red LED quadrants during the last 5 s of two 30 s activation period was significantly higher for SS33917>CsChrimson flies (right). (**E**) The probability of returning to the location where LED stimulation was terminated were measured as in *Figure 2*, but without airflow. See *Figure 7—figure supplement 1* for the time courses and other parameters. UpWiN drivers are shown together with SS49755 from the screen.

The online version of this article includes the following source data and figure supplement(s) for figure 7:

**Source data 1.** The values used for *Figure 7A*.

**Source data 2.** The values used for *Figure 7B*.

**Source data 3.** The values used for *Figure 7C*.

**Source data 4.** The values used for *Figure 7D*.

**Source data 5.** The values used for *Figure 7E*.

**Figure supplement 1.** UpWind Neuron (UpWiN) activation phenotypes without airflow.

**Figure supplement 1—source data 1.** The values used for *Figure 7—figure supplement 1*.

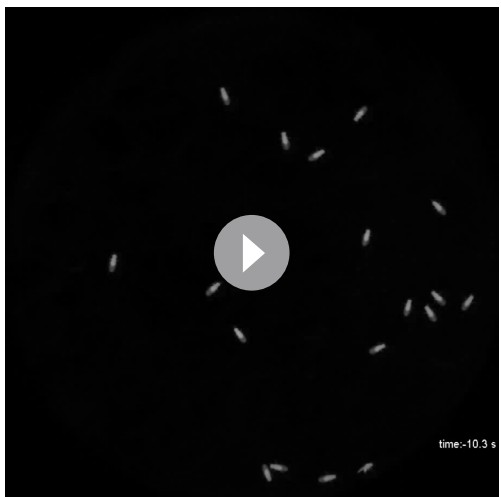

**Video 6.** Return phenotype induced by SMP357-362 activation. An example movie of SMP357-362 activation in SS49755>CsChrimson flies. The red square at the bottom right corner indicates the 10 s period when the red LED was turned on. The small circles indicate the centroid of flies and triangles indicate the orientation of flies. Trajectories of flies after turning off LED are shown as lines connecting centroids over time.

https://elifesciences.org/articles/85756/figures#video6

coordinated sequence of motor patterns that are deeply related to olfactory navigation, determination of turning direction and walking speed depended on fly's orientation to wind direction. Thus, our findings may mark an important transition point of the circuit, where abstract valence signals encoded by a population of neurons are evaluated and gradually transformed into concrete motor patterns.

## Memory valence and competing drives

Previous studies in the *Drosophila* MB have predicted the existence of a valence integration process. First, flies can create appetitive and aversive memories in parallel in different MB compartments after a single learning experience, and those memories compete over the behavioral choice with distinct time courses (*Aso et al., 2014b*; *Aso and Rubin, 2016*; *Das et al., 2014*; *Kaun et al., 2011*). Second, memory extinction (*Felsenberg et al., 2018*) and reversal learning (*McCurdy et al., 2021*) create a memory trace in an MB compartment, which neutralizes the effect of the original memory traces that persist in other MB compartments. Third, attraction and avoidance behaviors induced by photoactivation of multiple types of MBONs can be largely explained by the additive effects of individual activation (*Aso et al., 2014b*).

These studies support the 'valence-balance model', where learning-induced plasticity in the MB tips the balance of the valence signals of the MBON population (*Heisenberg, 2003*; *Hige, 2018*; *Modi et al., 2020*; *Owald and Waddell, 2015*). The mode of synaptic integration observed in the UpWiNs matches the expectation from this model. UpWiNs receive direct inhibitory and excitatory synaptic inputs from MBONs of appetitive and aversive memory compartments, respectively. When both presynaptic MBONs were activated, which mimicked the naive state (i.e. no depression in either of the MBONs), those inputs canceled each other, resulting in no net excitation (*Figure 4*). When plasticity was induced in the inhibitory appetitive-memory MBONs, which mimics appetitive memory formation, the odor response was enhanced (*Figure 5*). Thus, UpWiNs are able to decode the unbalanced activity of MBONs encoding opposing valence. Given the prevalence of convergent circuit motifs in the downstream circuits of the MB (*Li et al., 2020*), we predict that similar synaptic integration of those output neurons that signal the same or opposite stimulus valences controls other components of olfactory behaviors. Convergence of valence signals might also occur between the MB and LH, which is the other olfactory center parallel to the MB and is thought to mediate innate behavior. In fact, one MBON type sends its axon to the LH and causes learning-dependent modulation of the activity of food-odor-responding neurons (*Dolan et al., 2018*). UpWiNs also receive abundant input from the LH neurons, suggesting that UpWiNs also play an important role in integrating the innate and learned valances. The use of divergent and convergent pathways to process valence signals, like those we describe here, appears to be an evolutionarily conserved strategy that is observed, for example, in the vertebrate amygdala and its associated brain areas (*Tye, 2018*).

## UpWiNs and olfactory navigation

In addition to valence integration, UpWiNs play an important role in wind-guided behavior. Wind direction provides a critical cue for olfactory navigation in natural environments where odorants are propagated by the stream of airflow. Male moths have an astonishing ability to track the source of attractant pheromones emitted from females located over a mile away, and have been used as a model

for olfactory navigation (*Cardé, 2021*; *Kanzaki et al., 1994*; *Vergassola et al., 2007*). Male moths react to the intermittent plume of pheromone by series of cast-surge-cast actions (*Baker, 1990*). In a wind tunnel experiment, *Cadra cautella* moths began cross-wind casting following withdrawal of the pheromone plume. Upon contact with a single puff of pheromone, moths surged upwind after a delay of approximately 200 ms to reorient themselves. In our optogenetic experiments, activation of UpWiNs increased angular velocity with a similar time scale and biased turning direction toward the upwind direction (*Figure 6D–F*). In addition to promoting an upwind surge, UpWiNs activation increased the probability of returning to the location where activation was applied even after the cessation of both optogenetic activation and airflow (*Figure 7* and *Figure 7—figure supplement 1*). Therefore, we speculate that UpWiNs alone may be able to promote a series of cast-surge-cast reactions when flies navigate intermittent plumes of reward-predicting odors. Furthermore, as the third function, UpWiNs can promote release of dopamine in multiple MB compartments, presumably via converging connection with SMP108 which in turn feeds excitatory inputs to multiple DANs to instruct formation of second-order memories (*Yamada et al., 2023*). Interestingly, the patterns of DAN population responses to SMP108 or UpWiNs activation are similar to those observed when flies are walking toward vinegar in a virtual environment (*Zolin et al., 2021*). Together with the evidence of inputs from the lateral horn neurons, this may indicate that UpWiNs are also responsible for upwind locomotion to innately attractive odors and can be the causal source of action correlates in DANs. All three of these UpWiNs functions likely contribute to olfactory navigation in complex environments. Our study was limited to walking behaviors, and the role of UpWiNs in flight behaviors remains to be investigated. UpWiNs are also well positioned to influence internal state to promote selective processing of wind and olfactory stimuli.

Recent studies in *Drosophila* have provided insights into detailed neural circuit mechanisms of wind sensation and olfactory navigation. Both flying and walking flies turn upwind and increase locomotion speed when they encounter an attractive odor (*Álvarez-Salvado et al., 2018*; *Steck et al., 2012*; *van Breugel and Dickinson, 2014*). Airflow is detected by displacement of aristae and the Johnston organ sensory neurons (*Kamikouchi et al., 2006*; *Yorozu et al., 2009*), and left-right asymmetry is computed by the downstream neurons to represent wind direction in the central complex (*Matheson et al., 2022*; *Okubo et al., 2020*; *Suver et al., 2019*). Fictive appetitive and aversive training using optogenetic activation of DANs can promote and suppress the upwind locomotion, respectively (*Handler et al., 2019*), suggesting that retrieval of associative memory to drive behavior utilizes this same navigational strategy. Activation of a set of input neurons of the fan-shaped body (FB), which is a part of the central complex known as the navigation center of insects, can induce robust upwind locomotion (*Matheson et al., 2022*). The FB is one of the major downstream targets of MBONs (*Li et al., 2020*; *Scaplen et al., 2021*) while also receiving input from neurons representing wind directions (*Matheson et al., 2022*). Although these studies point to the importance of the central complex as the integration site of information about learned odor and wind direction, much remains to be learned about how the valence signals conveyed by MBONs influence upwind locomotion.

Based on the EM connectome data, SMP108 appears to be the most prominent neuron postsynaptic to UpWiNs. Activation of SMP108 was able to promote upwind locomotion, but the details of behavioral response differed from UpWiNs activation (*Figure 2*), and blocking SMP108 did not affect retrieval of appetitive memory (*Yamada et al., 2023*). Therefore, UpWiNs may evoke upwind behavior through other downstream cells. FB6D, FB6I, and FB6T appear to be other major downstream neurons of UpWiNs (*Figure 3C*). The top three downstream neurons of FB6D are hDeltaF, hDeltaC, and hDeltaK. hDeltaC is the columnar cell type of the FB that is known to integrate wind-directional cues and information of innately attractive odor to promote upwind behavior (*Matheson et al., 2022*). Our screening also identified that coactivation of hDeltaB, hDeltaD, and hDeltaE can promote robust upwind locomotion (*Figure 2* and *Figure 2—figure supplements 1 and 3*). Thus, these hDelta cell types likely function together to regulate wind-directed locomotion. EM-connectome-guided follow-up studies on other cell types with significant upwind/downwind phenotypes (*Figure 2*) will help generate a comprehensive understanding of olfactory navigation circuits.

## Compartment-specific contribution to anemotaxis

One highlight of our study is the finding that memories stored in different MB compartments use different behavioral strategies during retrieval. Although gaining a full description of those behavioral

strategies is beyond the scope of the present study, we can speculate on the potential biological significance of the differential contributions to anemotaxis—movement in response to air currents—behaviors by MB compartments.

First, the difference in the type of memory stored in different compartments is likely to be a key factor. Based on the circuit connectivity and behavioral data, UpWiNs are responsible for upwind locomotion driven by the memory stored in the α1 compartment. Compared to other DANs for appetitive memory, the DANs in the α1 only weakly respond to sugar (*Yamagata et al., 2015*), and write a memory slowly even when optogenetically activated (*Aso and Rubin, 2016*; *Yamada et al., 2023*). Once formed with repetitive training, the memory in the α1 lasts over a day and is most resistant to extinction and decay over time (*Aso and Rubin, 2016*; *Huetteroth et al., 2015*; *Ichinose et al., 2015*; *Yamada et al., 2023*; *Yamagata et al., 2015*). These features collectively indicate that flies undergo wind-guided olfactory navigation only when they expect a robust/reliable reward (*Figure 1E*).

Second, the MB must operate with different downstream circuits in adults and larva. Holometabolan insects undergo complete metamorphosis by which body structures of larvae abruptly develop into adult's form through pupal stage. In *Drosophila*, behavioral components of olfactory navigation and relevant neural circuits also undergo striking changes through metamorphosis. *Drosophila* larvae hatch from eggs with already developed circuits of olfaction and an MB that consists of ~70 mature KCs, 7 DANs, and 24 MBONs (*Eichler et al., 2017*). The first instar larval MB circuit is numerically simpler than that of the adult but can support associative learning (*Pauls et al., 2010*). Larval *Drosophila* perform innate and memory-based olfactory navigation by modulating rate of head casting and reorientation based on concentration gradient of odors measured over time (*Fishilevich et al., 2005*; *Saumweber et al., 2018*). Although larvae can sense wind and use it to avoid aversive odors (*Jovanic et al., 2019*), they do not use the wind direction to localize the source of attractive odors as adults do. The adult airflow-sensing organ (i.e. aristae), relevant neural circuits such as the central complex, legs, and wings all develop during metamorphosis. Therefore, the metamorphing MB circuit must adopt new interacting partners to transform stored memories into adult-specific anemotaxis behaviors.

The EM connectome of larval and adult MB circuits revealed many cell types with similar morphology, which are in some cases labeled by the same genetic driver lines (*Aso et al., 2014a*; *Eichler et al., 2017*; *Li et al., 2020*). To unambiguously match larval and adult cell types, a recent study followed full developmental trajectories of larval MB cell types into the adult MB by immobilizing expression patterns of genetic driver lines (*Truman et al., 2023*). Intriguingly, a large fraction of MBONs and DANs survive through metamorphosis and become a part of the adult MB circuit. For instance, among 17 types of larval MBONs examined, 10 types developed into adult MBONs. These larval-origin MBONs arborize their dendrites in the γ or β' lobes. Our experiments indicated that appetitive memories in γ4 and γ5β'2a compartments can bias the choice between quadrants filled with CS+ and CS− odors but do not promote walking toward upwind (*Figure 1*). This could be because the γ lobe stores both olfactory and visual memories (*Vogt et al., 2016*; *Vogt et al., 2014*); walking upwind does not help get closer to visual cues associated with reward. On the other hand, α/β KCs, MBON-α1, MBON-α3, MBON-β1>α are adult specific cell types. Notably, MBON-β1>α innervates the α lobe and is directly connected with MBON-α1 and MBON-α3, suggesting that UpWiNs may integrate information from the β1, α1, and α3. These anatomical observations suggest that the adult-specific output pathways of MB may be dedicated to anemotaxis. In naturalistic olfactory conditioning with sugar reward, flies form parallel appetitive memories in compartments of both larval-origin and adult-specific MBONs. Future EM-connectome-guided studies will elucidate how the adult MB integrates parallel memories to synthesize navigational strategies by blending anemotactic and other behavioral components.

## Materials and methods
### Fly strains
*Drosophila melanogaster* strains were reared at 22°C and 60% humidity on standard cornmeal food in a 12:12 hr light:dark cycle. 4- to 10-day-old adult females were used 2–4 days after sorting them on a Peltier cold plate. For flies expressing CsChrimson (*Klapoetke et al., 2014*), the food was supplemented with retinal (0.2 mM all-trans-retinal prior to eclosion and then 0.4 mM). Driver and effector lines are listed in Key resources table and genotypes used by each figure are listed below. The new

collection of split-GAL4 drivers was designed based on confocal image databases (http://flweb.janelia.org) (*Jenett et al., 2012*), and screening expression patterns of p65ADZp and ZpGAL4DBD combinations are described previously (*Aso et al., 2014a*; *Pfeiffer et al., 2010*) and in the accompanying article (*Shuai et al., 2023*). Confocal stacks of new split-GAL4 driver lines used in this study are available at http://www.janelia.org/split-gal4.

## Olfactory conditioning

Olfactory conditioning was performed as previously described (*Aso and Rubin, 2016*). Groups of approximately 20 females of 4- to 10-day post-eclosion were trained and tested using the modified four-field olfactory arena (*Aso and Rubin, 2016*; *Pettersson, 1970*) equipped with a 627 nm LED board (34.9 µW/mm$^2$ at the position of the flies) and odor mixers. The flow rate of input air from each of the four arms was maintained at 100 mL/min throughout the experiments by mass flow controllers, and air was pulled from the central hole at 400 mL/min. Odors were delivered to the arena by switching the direction of airflow to the tubes containing diluted odors using solenoid valves. The odors were diluted in paraffin oil: PA (1:10,000) and EL (1:10,000). Sugar conditioning was performed by using tubes with sucrose absorbed Whatman 3 MM paper that was dried before use as previously described (*Krashes and Waddell, 2008*; *Liu et al., 2012*). For conditioning with optogenetic activation of DANs, 60 s of odor was paired with 30 times 1 s of red LED light with 1 s gaps. LED pulses started 5 s after the opening of odor valves. Before conditioning, flies were starved for 40–48 hr on 1% agar. Videography was performed at 30 frames per second and analyzed using Flytracker (https://github.com/kristinbranson/FlyTracker; *Branson, 2023*) or Fiji. For experiments using 1 day memory retention, flies were kept in agar vials at 21°C after first-order conditioning. For testing olfactory memories, the distribution of flies in the four quadrants was measured for 60 s. The performance index (PI) is defined as a mean of [(number of flies in the two diagonal quadrants filled with odor one) – (number of flies in other two quadrants filled with odor two or air)]/(total number of flies) during the final 30 s of the 60 s test period. The average PI of reciprocal experiments is shown in figures to cancel out potential position bias and innate odor preference.

## Airflow response

For testing airflow directional response, each fly's distance from center ($r_i$) was measured. The radius of the arena, $r_{arena}$, was 50 mm. Because of the circular shape of the arena, the area of particular r bin is larger as r increases. For instance, the area of 40 < r < 50 mm is nine times larger than the area of 0 < r < 10 mm. When flies distribute randomly in the arena, the mean $r_i$ is 1/sqrt(2). To normalize this area difference we used the square of ($r_i/r_{arena}$) as an area-normalized distance from the center index. To calculate upwind displacement, the mean of area-normalized distance from center at each time point in each movie was subtracted by the area-normalized distance at the onset of activating illumination or odor presentation. To compensate for the delay between the switch of solenoid valves and delivery of the odor (~2 s) as well as the time to fill the arena with odorized air (~3 s), the onset of odor was taken to be 3.5 s after the switch of solenoid valves. For analysis of individual trajectories, only flies that were more than 3 mm away from the edge of the arena were analyzed. Trajectories with too abrupt changes of angle (more than 180 degree) or position (more than 5 mm) in one frame were considered as tracking errors and excluded from the analysis. The direction toward the center of the arena, where suction tubing is connected, was designed as ±180 degrees relative to the upwind direction. For analyzing the influence of initial orientation on directional turning and forward walking speed, subsets of trajectories were analyzed by grouping them into ±30 degree bins of initial angle.

## Electrophysiology

Electrophysiological experiments were performed as previously described (*Yamada et al., 2023*). Briefly, flies were collected on the day of eclosion and kept in the dark on all-trans-retinal food (0.5 mM) until experiments for 48–72 hr. The patch pipettes (6–7 MΩ) were filled with the pipette solution containing (in mM): L-potassium aspartate, 140; HEPES, 10; EGTA, 1.1; CaCl$_2$, 0.1; Mg-ATP, 4; Na-GTP, 0.5 with pH adjusted to 7.3 with KOH (265 mOsm). The preparation was continuously perfused with saline containing (in mM): NaCl, 103; KCl, 3; CaCl$_2$, 1.5; MgCl$_2$, 4; NaHCO$_3$, 26; *N*-tris(hydroxymethyl) methyl-2-aminoethane-sulfonic acid, 5; NaH$_2$PO$_4$, 1; trehalose, 10; glucose, 10 (pH 7.3 when bubbled with 95% O$_2$ and 5% CO$_2$, 275 mOsm). UpWiNs were visually identified by fluorescence

signals expressed by specific drivers. Whole-cell recordings were made using the Axon MultiClamp 700B amplifier (Molecular Devices). Cells were held at around −60 mV by injecting hyperpolarizing current, which was typically less than 10 pA. Signals were low-pass filtered at 5 kHz and digitized at 10 kHz before being acquired and analyzed by custom MATLAB scripts (MathWorks). Subthreshold odor responses were quantified by averaging the mean depolarization above the baseline during 0–1.2 s after odor onset. Saturated head-space vapors of odors were presented to flies after 1% air dilution using a custom odor delivery system. 625 nm LEDs were used to deliver photostimulation at 17 mW/mm$^2$ through the objective lens.

## Calcium imaging

All experiments were performed on female flies, 3–7 days after eclosion. Brains were dissected in a saline bath (103 mM NaCl, 3 mM KCl, 2 mM CaCl$_2$, 4 mM MgCl$_2$, 26 mM NaHCO$_3$, 1 mM NaH$_2$PO$_4$, 8 mM trehalose, 10 mM glucose, 5 mM TES, bubbled with 95% O$_2$/5% CO$_2$). After dissection, the brain was positioned anterior side up on a coverslip in a Sylgard dish submerged in 3 mL saline at 20°C. The sample was imaged with a resonant scanning two-photon microscope with near-infrared excitation (920 nm, Spectra-Physics, INSIGHT DS DUAL) and a 25× objective (Nikon MRD77225 25XW). The microscope was controlled using ScanImage 2016 (Vidrio Technologies). Images were acquired over a 231 μm × 231 μm × 42 μm volume with a step size at 2 μm. The field of view included 512×512 pixel resolution taken at approximately 1.07 Hz frame rate. The excitation power during imaging was 19 mW (*Figure 4C*) or 12 mW (*Figure 4—figure supplement 1*).

For photostimulation, the light-gated ion channel Chrimson88 (*Klapoetke et al., 2014*) was activated with a 660 nm LED (M660L3 Thorlabs) coupled to a digital micromirror device (Texas Instruments DLPC300 Light Crafter) and combined with the imaging path using an FF757-DiO1 dichroic (Semrock). On the emission side, the primary dichroic was Di02-R635 (Semrock), the detection arm dichroic was 565DCXR (Chroma), and the emission filters were FF03-525/50 and FF01-625/90 (Semrock). Photostimulation occurred over a 1 s period at a 12 μW/mm$^2$ (*Figure 4C*) or 7.8 μW/mm$^2$ (*Figure 4—figure supplement 1*) intensity over nine consecutive trials interspersed by a 30 s period. The light intensity was measured using a Thorlabs S170C power sensor.

When quantifying the GCaMP fluorescence, regions of interest (ROIs) corresponding to MB compartments were drawn using custom python scripts on images showing the maximum intensity over time. Mean intensity changes within these ROIs were measured in the time series images. Final intensity measurements subtracted a background ROI that was drawn in a region with no fluorescence. Baseline fluorescence is the mean fluorescence over a 5 s time period before stimulation started. The ΔF was then divided by baseline to normalize the signal (ΔF/F). The mean responses from the nine trials were calculated for each animal (5–11 samples per driver). Although RNAi should knock down GCaMP6s expression in UpWiNs expressing Chrimson88-tdTomato in *Figure 4—figure supplement 1*, voxels including neurons expressing red fluorescence (tdTomato) were excluded from the analysis. This exclusion was performed by manually selecting a minimum threshold that identified red fluorescent regions corresponding to tdTomato expressing neurons. Voxels in the red channel above this threshold were excluded in the green channel measuring GCaMP6s fluorescence.

## Analysis of connectivity and morphology

The information was retrieved from neuPrint (https://neuprint.janelia.org/) hosting the 'hemibrain' dataset (*Scheffer et al., 2020*), which is a publicly accessible web site (https://doi.org/10.25378/janelia.12818645.v1). For cell types, we cited cell type assignments reported in *Scheffer et al., 2020*. Only connections of the cells in the right hemisphere were used due to incomplete connectivity in the left hemisphere (*Zheng et al., 2018*). Connectivity data was then imported into Cytoscape (https://cytoscape.org/) for generating circuit diagrams that were edited using Adobe Illustrator. The 3D renderings of neurons presented were generated using the visualization tools of NeuTu (*Zhao et al., 2018*) or VVD viewer (https://github.com/takashi310/VVD_Viewer; *Kawase, 2014*; *Wan et al., 2012*). Morphological similarity of individual neurons in SS33917 driver was performed by NBLAST (*Costa et al., 2016*).

## Immunohistochemistry

Brains and ventral nerve cord of 4- to 10-day-old females were dissected, fixed, and immunolabeled as previously described using the antibodies listed in Key resources table (*Aso et al., 2014a*; *Nern et al., 2015*).

## Statistics

Statistical comparisons were performed on GraphPad Prism or MATLAB using the Kruskal-Wallis test followed by Dunn's post-test for multiple comparison, t-tests, or two-way ANOVA followed by Tukey's post hoc multiple comparisons test as designated in figure legends.

## Detailed fly genotypes used by figures

| Figure | Genotype |
| --- | --- |
| *Figure 1D–F* | *w/w, 20xUAS-CsChrimson-mVenus attP18;;+/Gr64f-split-GAL4 (SS87269)* |
| *Figure 1H–L*<br>*Figure 1—figure supplement 1* | *w/w, 20xUAS-CsChrimson-mVenus attP18;;+/MB043C-split-GAL4*<br>*w/w, 20xUAS-CsChrimson-mVenus attP18;+/MB213B-split-GAL4*<br>*w/w, 20xUAS-CsChrimson-mVenus attP18;;+/MB312C-split-GAL4*<br>*w/w, 20xUAS-CsChrimson-mVenus attP18;MB109B/MB315C-split-GAL4*<br>*w/w, 20xUAS-CsChrimson-mVenus attP18;+/ Empty-split-GAL4* |
| *Figure 2*<br>*Figure 2—figure supplements 1 and 2* | *w/w, 20xUAS-CsChrimson-mVenus attP18;Split-GAL4/+* |
| *Figure 3A* | *w/w, 20xUAS-CsChrimson-mVenus attP18;SS33917-Split-GAL4/+* |
| *Figure 3—figure supplement 2* | *pBPhsFlp2::PEST in attP3;; pJFRC201-10XUAS-FRT>STOP > FRT-myr::smGFP-HA in VK0005, pJFRC240-10XUAS- FRT>STOP > FRT-myr::smGFP-V5-THS-10XUAS-FRT>STOP > FRT-myr::smGFP-FLAG in su(Hw)attP1/SS33917-split-GAL4* |
| *Figure 4A*<br>*Figure 4B*<br>*Figure 4C* | *LexAop2-Syn21-opGCaMP6s (JK16F), LexAop2-Syn21-opGCaMP6s (SuHwattP8), 10XUAS-Syn21-Chrimson88-tdT-3.1 (attP18)/+; R64A11-LexAp65 (JK73A)/MB082C*<br>*LexAop2-Syn21-opGCaMP6s (JK16F), LexAop2-Syn21-opGCaMP6s (SuHwattP8), 10XUAS-Syn21-Chrimson88-tdT-3.1 (attP18)/+; R64A11-LexAp65 (JK73A)/MB310C*<br>*LexAop2-Syn21-opGCaMP6s (JK16F), LexAop2-Syn21-opGCaMP6s (SuHwattp8), 10xUAS-Syn21-Chrimson88-tdT-3.1 (attp18); G0239-GAL4/ G0239-GAL4; R64A11-LexAp65 (JK73A)/+*<br>*LexAop2-Syn21-opGCaMP6s (JK16F), LexAop2-Syn21-opGCaMP6s (SuHwattp8), 10xUAS-Syn21-Chrimson88-tdT-3.1 (attp18); G0239-GAL4/ G0239-GAL4; R64A11-LexAp65 (JK73A)/MB310C*<br>*LexAop2-Syn21-opGCaMP6s (JK16F), LexAop2-Syn21-opGCaMP6s (SuHwattp8), 10xUAS-Syn21-Chrimson88-tdT-3.1 (attp18); G0239-GAL4/+; R64A11-LexAp65 (JK73A)/MB310C* |
| *Figure 4—figure supplement 1* | *LexAop2-Syn21-opGCaMP6s in su(Hw)attP8,10XUAS-Syn21-Chrimson88-tdT-3.1 in attP18; 10XUAS-LexAp65-DBD2-RNAi (VK2)/; 64A11-LexAp65 JK73A/ SS67249* |
| *Figure 5* | *13XLexAop2-IVS-p10-ChrimsonR-mVenus (attP18); 58E02-LexAp65 (attP40)/ss67249-split1; pJFRC28-10XUAS-IVS-GFP-p10 (SuHwattP1) / ss67249-split2* |
| *Figure 5—figure supplement 1* | *pBPhsFlp2::PEST in attP3;; pJFRC201-10XUAS-FRT>STOP > FRT-myr::smGFP-HA in VK0005, pJFRC240-10XUAS- FRT>STOP > FRT-myr::smGFP-V5-THS-10XUAS-FRT>STOP > FRT-myr::smGFP-FLAG in su(Hw)attP1/SS67249-split-GAL4* |
| *Figure 5—figure supplement 2* | *13XLexAop2-IVS-p10-ChrimsonR-mVenus (attP18); 58E02-LexAp65 (attP40)/ss67249-split1; pJFRC28-10XUAS-IVS-GFP-p10 (SuHwattP1) / ss67249-split2* |
| *Figure 6* | *w/w, 20xUAS-CsChrimson-mVenus attP18;SS33917-Split-GAL4/+w/w, 20xUAS-CsChrimson-mVenus attP18;SS33918-Split-GAL4/+w/w, 20xUAS-CsChrimson-mVenus attP18;MB077B-Split-GAL4/+w/w, 20xUAS-CsChrimson-mVenus attP18;Empty-Split-GAL4/+* |

*Continued on next page*

*Continued*

| Figure | Genotype |
|---|---|
| *Figure 6—figure supplement 1* | *w/w, 20xUAS-CsChrimson-mVenus attP18;Split-GAL4/+* |
| *Figure 7A–B* | *w/+;SS33917-split-GAL4/+w/+; SS33917-split-GAL4UAS-TNT (II) w/+; Empty-split-GAL4UAS-TNT (II)* |
| *Figure 7C* | *w/w;VT007746-p65ADZp in attP40/20xUAS-Shbire-p10 in VK00005 w/w;R64A11-ZpGAL4DBD in attP2/20xUAS-Shbire-p10 in VK00005 w/w;SS33917(VT007746-p65ADZp in attP40; R64A11-ZpGAL4DBD in attP2)/20xUAS-Shbire-p10 in VK00005 w/w;Empty-split-GAL4/20xUAS-Shbire-p10 in VK00005* |
| *Figure 7D* | *w/w, 20xUAS-CsChrimson-mVenus attP18;SS33917/+w/w, 20xUAS-CsChrimson-mVenus attP18;Empty-split-GAL4 /+* |
| *Figure 7E* | *w/w, 20xUAS-CsChrimson-mVenus attP18;SS33917/+w/w, 20xUAS-CsChrimson-mVenus attP18;SS33918/+w/w, 20xUAS-CsChrimson-mVenus attP18;SS49755/+w/w, 20xUAS-CsChrimson-mVenus attP18;Empty-split-GAL4 /+* |
| *Figure 7—figure supplement 1* | *w/w, 20xUAS-CsChrimson-mVenus attP18;SS33917/+w/w, 20xUAS-CsChrimson-mVenus attP18;SS33918/+w/w, 20xUAS-CsChrimson-mVenus attP18;SS49755/+w/w, 20xUAS-CsChrimson-mVenus attP18;Gr64f-GAL4;Gr64f-GAL4/+w/w, 20xUAS-CsChrimson-mVenus attP18;Empty-split-GAL4 /+* |

## Acknowledgements

We thank Gerald M Rubin, Glenn Turner, Ashok Litwin-Kumar, Daisuke Hattori, Gown Turvo, Huai-ti Lin, and members of the YA and TH labs for valuable discussion and comments on the manuscript. We thank Feng Li, Marisa Dreher, Christina Christoforou, Kari Close, and the members of Janelia FlyEM, Flylight, fly facility, Project Technical Resource, and scientific computing for technical support. DY was supported by Toyobo Biotechnology Foundation Postdoctoral Fellowship and Japan Society for the Promotion of Science Overseas Research Fellowship. YA was supported by HHMI. TH was supported by NIH (R01DC018874), NSF (2034783), BSF (2019026), and UNC Junior Faculty Development Award.

## Additional information

### Funding

| Funder | Grant reference number | Author |
|---|---|---|
| Howard Hughes Medical Institute | | Yoshinori Aso |
| National Institutes of Health | R01DC018874 | Toshihide Hige |
| National Science Foundation | 2034783 | Toshihide Hige |
| US-Israel Binational Science Foundation | 2019026 | Toshihide Hige |
| UNC Junior Faculty Development Award | | Toshihide Hige |
| Japan Society for the Promotion of Science | Overseas Research Fellowship | Daichi Yamada |
| Toyobo Biotechnology Foundation | Postdoctoral Fellowship | Daichi Yamada |

The funders had no role in study design, data collection and interpretation, or the decision to submit the work for publication.

## Author contributions
Yoshinori Aso, Conceptualization, Resources, Data curation, Software, Formal analysis, Supervision, Funding acquisition, Validation, Investigation, Visualization, Methodology, Writing – original draft, Project administration, Writing – review and editing; Daichi Yamada, Daniel Bushey, Formal analysis, Investigation, Writing – review and editing; Karen L Hibbard, Resources; Megan Sammons, Data curation, Investigation; Hideo Otsuna, Software, Formal analysis; Yichun Shuai, Data curation, Investigation, Writing – review and editing; Toshihide Hige, Conceptualization, Data curation, Software, Formal analysis, Supervision, Funding acquisition, Investigation, Visualization, Writing – original draft, Project administration, Writing – review and editing

## Author ORCIDs
Yoshinori Aso ⓘ https://orcid.org/0000-0002-2939-1688
Daniel Bushey ⓘ http://orcid.org/0000-0001-9258-6579
Karen L Hibbard ⓘ http://orcid.org/0000-0002-2001-6099
Megan Sammons ⓘ http://orcid.org/0000-0003-4516-5928
Hideo Otsuna ⓘ http://orcid.org/0000-0002-2107-8881
Toshihide Hige ⓘ https://orcid.org/0000-0002-0007-3192

## Decision letter and Author response
Decision letter https://doi.org/10.7554/eLife.85756.sa1
Author response https://doi.org/10.7554/eLife.85756.sa2

---

# Additional files

## Supplementary files
• MDAR checklist

## Data availability
The confocal images of expression patterns are available online (http://www.janelia.org/split-gal4). The values used for figures are summarized in Source Data. The design files of the olfactory arena were initially described in Aso & Rubin 2016 and are freely available at https://www.janelia.org/open-science/four-way-olfactometer-arena-fruit-flies and flintbox (https://hhmi.flintbox.com/technologies/c65b2ddd-3cc1-44d9-a95d-73e08723f724) with the license agreement.

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

# Appendix 1

## Appendix 1—key resources table

| Reagent type (species) or resource | Designation | Source or reference | Identifiers | Additional information |
|---|---|---|---|---|
| Strain, strain background (*Drosophila melanogaster*) | 20xUAS-CsChrimson-mVenus attP18 | *Klapoetke et al., 2014*; PMID:24509633 | N.A. | |
| Strain, strain background (*Drosophila melanogaster*) | 10XUAS-Chrimson88-tdTomato attP1 | *Klapoetke et al., 2014*; PMID:24509633 | N.A. | |
| Strain, strain background (*Drosophila melanogaster*) | 13XLexAop2-IVS-ChrimsonR-mVenus-p10 attP18 | Vivek Jayaraman | N.A. | |
| Strain, strain background (*Drosophila melanogaster*) | 20XUAS-syn21-mScarlet-opt-p10 su(Hw)attp8 | Glenn Turner | N.A. | |
| Strain, strain background (*Drosophila melanogaster*) | pJFRC200-10xUAS-IVS-myr::smGFP-HA in attP18 | *Nern et al., 2015*; PMID:25964354 | N.A. | |
| Strain, strain background (*Drosophila melanogaster*) | pJFRC225-5xUAS-IVS-myr::smGFP-FLAG in VK00005 | *Nern et al., 2015*; PMID:25964354 | N.A. | |
| Strain, strain background (*Drosophila melanogaster*) | pBPhsFlp2::PEST in attP3 | *Nern et al., 2015*; PMID:25964354 | N.A. | |
| Strain, strain background (*Drosophila melanogaster*) | pJFRC201-10XUAS-FRT>STOP > FRT-myr::smGFP-HA in VK0005 | *Nern et al., 2015*; PMID:25964354 | N.A. | |
| Strain, strain background (*Drosophila melanogaster*) | pJFRC240-10XUAS-FRT>STOP > FRT-myr::smGFP-V5-THS-10XUAS-FRT>STOP > FRT-myr::smGFP-FLAG_in_su(Hw)attP1 | *Nern et al., 2015*; PMID:25964354 | N.A. | |
| Strain, strain background (*Drosophila melanogaster*) | MB043-split-LexA | *Shuai et al., 2023*; This paper | N.A. | Available from Aso lab |
| Strain, strain background (*Drosophila melanogaster*) | empty-split-GAL4 (p65ADZp attP40, ZpGAL4DBD attP2) | *Seeds et al., 2014*; PMID:25139955 | N.A. | |
| Strain, strain background (*Drosophila melanogaster*) | MB001B (R10H10-p65ADZp in attP40 and R55A07ZpGAL4DBD in attP2) | *Shuai et al., 2023*; This paper | N.A. | Available from Aso lab |
| Strain, strain background (*Drosophila melanogaster*) | MB002B (R12C11-p65ADZp in attP40 and R14C08ZpGAL4DBD in attP2) | *Aso et al., 2014a* eLife, DOI:10.7554/eLife.04577 | N.A. | |
| Strain, strain background (*Drosophila melanogaster*) | MB011C (R14C08-p65ADZp in VK00027 and R15B01ZpGAL4DBD in attP2) | *Aso et al., 2014a* eLife, DOI:10.7554/eLife.04577 | N.A. | |
| Strain, strain background (*Drosophila melanogaster*) | MB018B (R20G03-p65ADZp in attP40 and R19F09ZpGAL4DBD in attP2) | *Aso et al., 2014a* eLife, DOI:10.7554/eLife.04577 | N.A. | |
| Strain, strain background (*Drosophila melanogaster*) | MB022B (R24E06-p65ADZp in attP40 and Tdc2ZpGAL4DBD in attP2) | *Aso et al., 2014a* eLife, DOI:10.7554/eLife.04577 | N.A. | |
| Strain, strain background (*Drosophila melanogaster*) | MB025B (R24E12-p65ADZp in attP40 and R52H01ZpGAL4DBD in attP2) | *Aso et al., 2014a* eLife, DOI:10.7554/eLife.04577 | N.A. | |
| Strain, strain background (*Drosophila melanogaster*) | MB027B (R24H08-p65ADZp in attP40 and R53F03ZpGAL4DBD in attP2) | *Aso et al., 2014a* eLife, DOI:10.7554/eLife.04577 | N.A. | |
| Strain, strain background (*Drosophila melanogaster*) | MB029B (R30G08-p65ADZp in attP40 and R11A03ZpGAL4DBD in attP2) | *Shuai et al., 2023*; This paper | N.A. | Available from Aso lab |
| Strain, strain background (*Drosophila melanogaster*) | MB032B (R30G08-p65ADZp in attP40 and THZpGAL4DBD in attP2) | *Aso et al., 2014a* eLife, DOI:10.7554/eLife.04577 | N.A. | |
| Strain, strain background (*Drosophila melanogaster*) | LH2456 (R38D01-p65ADZp in attP40 and R77F05ZpGAL4DBD in attP2) | *Dolan et al., 2019* eLife, DOI:10.7554/eLife.43079 | N.A. | |
| Strain, strain background (*Drosophila melanogaster*) | MB043C (R58E02-p65ADZp in VK00027 and R32D11ZpGAL4DBD in attP2) | *Aso et al., 2014a* eLife, DOI:10.7554/eLife.04577 | N.A. | |
| Strain, strain background (*Drosophila melanogaster*) | MB077B (R25D01-p65ADZp in attP40 and R19F09ZpGAL4DBD in attP2) | *Aso et al., 2014a* eLife, DOI:10.7554/eLife.04577 | N.A. | |
| Strain, strain background (*Drosophila melanogaster*) | MB080C (R33E02-p65ADZp in attP40 and R50A05ZpGAL4DBD in attP2) | *Aso et al., 2014a* eLife, DOI:10.7554/eLife.04577 | N.A. | |
| Strain, strain background (*Drosophila melanogaster*) | MB082C (R40B08-p65ADZp in VK00027 and R23C06ZpGAL4DBD in attP2) | *Aso et al., 2014a* eLife, DOI:10.7554/eLife.04577 | N.A. | |
| Strain, strain background (*Drosophila melanogaster*) | MB083C (R52G04-p65ADZp in VK00027 and R94B10ZpGAL4DBD in attP2) | *Aso et al., 2014a* eLife, DOI:10.7554/eLife.04577 | N.A. | |

*Appendix 1 Continued on next page*

*Appendix 1 Continued*

| Reagent type (species) or resource | Designation | Source or reference | Identifiers | Additional information |
|---|---|---|---|---|
| Strain, strain background (*Drosophila melanogaster*) | MB109B (R76F05-p65ADZp in attP40 and R23C12ZpGAL4DBD in attP2) | *Aso et al., 2014a* eLife, DOI:10.7554/eLife.04577 | N.A. | |
| Strain, strain background (*Drosophila melanogaster*) | MB112C (R93D10-p65ADZp in VK00027 and R13F04ZpGAL4DBD in attP2) | *Aso et al., 2014a* eLife, DOI:10.7554/eLife.04577 | N.A. | |
| Strain, strain background (*Drosophila melanogaster*) | MB213B (R76F05-p65ADZp in attP40 and R32G08ZpGAL4DBD in attP2) | *Aso et al., 2014a* eLife, DOI:10.7554/eLife.04577 | N.A. | |
| Strain, strain background (*Drosophila melanogaster*) | MB296B (R15B01-p65ADZp in attP40 and R26F01ZpGAL4DBD in attP2) | *Aso et al., 2014a* eLife, DOI:10.7554/eLife.04577 | N.A. | |
| Strain, strain background (*Drosophila melanogaster*) | MB310C (R52G04-p65ADZp in VK00027 and R17C11ZpGAL4DBD in attP2) | *Aso et al., 2014a* eLife, DOI:10.7554/eLife.04577 | N.A. | |
| Strain, strain background (*Drosophila melanogaster*) | MB315C (R58E02-p65ADZp in attP40 and R48H11ZpGAL4DBD in attP2) | *Aso et al., 2014a* eLife, DOI:10.7554/eLife.04577 | N.A. | |
| Strain, strain background (*Drosophila melanogaster*) | MB390B (R19B06-p65ADZp in attP40 and R59G08ZpGAL4DBD in attP2) | *Shuai et al., 2023*; This paper | N.A. | Available from Aso lab |
| Strain, strain background (*Drosophila melanogaster*) | MB433B (R30E08-p65ADZp in attP40 and R11C07ZpGAL4DBD in attP2) | *Aso et al., 2014a* eLife, DOI:10.7554/eLife.04577 | N.A. | |
| Strain, strain background (*Drosophila melanogaster*) | MB434B (R30E08-p65ADZp in attP40 and R53C10ZpGAL4DBD in attP2) | *Aso et al., 2014a* eLife, DOI:10.7554/eLife.04577 | N.A. | |
| Strain, strain background (*Drosophila melanogaster*) | MB441B (R30G08-p65ADZp in attP40 and R48B03ZpGAL4DBD in attP2) | *Aso et al., 2014a* eLife, DOI:10.7554/eLife.04577 | N.A. | |
| Strain, strain background (*Drosophila melanogaster*) | MB555B (R73F07-p65ADZp in attP40 and R12D12ZpGAL4DBD in attP2) | *Shuai et al., 2023*; This paper | N.A. | Available from Aso lab |
| Strain, strain background (*Drosophila melanogaster*) | MB630B (VT026773-p65ADZp in attP40 and R72B05ZpGAL4DBD in attP2) | *Aso and Rubin, 2016* eLife, DOI:10.7554/eLife.16135 | N.A. | |
| Strain, strain background (*Drosophila melanogaster*) | SS00096 (R19G02-p65ADZp in attP40 and R70G12ZpGAL4DBD in attP2) | *Turner-Evans et al., 2017* Elife, DOI:10.7554/eLife.23496 | N.A. | |
| Strain, strain background (*Drosophila melanogaster*) | SS00504 (R25C01-p65ADZp in attP40 and R54H01ZpGAL4DBD in attP2) | *Shuai et al., 2023*; This paper | N.A. | Available from Aso lab |
| Strain, strain background (*Drosophila melanogaster*) | SS00543 (R33E06-p65ADZp in VK00027 and R89G09ZpGAL4DBD in attP2) | *Shuai et al., 2023*; This paper | N.A. | Available from Aso lab |
| Strain, strain background (*Drosophila melanogaster*) | SS00550 (R37A12-p65ADZp in attP40 and R27H08ZpGAL4DBD in attP2) | *Shuai et al., 2023*; This paper | N.A. | Available from Aso lab |
| Strain, strain background (*Drosophila melanogaster*) | SS00561 (R38E07-p65ADZp in attP40 and R55B01ZpGAL4DBD in attP2) | *Shuai et al., 2023*; This paper | N.A. | Available from Aso lab |
| Strain, strain background (*Drosophila melanogaster*) | SS00581 (R48H12-p65ADZp in attP40 and R41B07ZpGAL4DBD in attP2) | *Shuai et al., 2023*; This paper | N.A. | Available from Aso lab |
| Strain, strain background (*Drosophila melanogaster*) | SS00623 (R76F12-p65ADZp in attP40 and R33E06ZpGAL4DBD in attP2) | *Shuai et al., 2023*; This paper | N.A. | Available from Aso lab |
| Strain, strain background (*Drosophila melanogaster*) | SS01126 (R14C08-p65ADZp in attP40 and VT 037491ZpGAL4DBD in attP2) | *Shuai et al., 2023*; This paper | N.A. | Available from Aso lab |
| Strain, strain background (*Drosophila melanogaster*) | SS01227 (R14C08-p65ADZp in attP40 and R11F12ZpGAL4DBD in attP2) | *Shuai et al., 2023*; This paper | N.A. | Available from Aso lab |
| Strain, strain background (*Drosophila melanogaster*) | SS01262 (VT 029592-p65ADZp in attP40 and R19B06ZpGAL4DBD in attP2) | *Shuai et al., 2023*; This paper | N.A. | Available from Aso lab |
| Strain, strain background (*Drosophila melanogaster*) | SS01319 (VT 029592-p65ADZp in attP40 and R33H11ZpGAL4DBD in attP2) | *Shuai et al., 2023*; This paper | N.A. | Available from Aso lab |
| Strain, strain background (*Drosophila melanogaster*) | SS01126 (R14C08-p65ADZp in attP40 and VT 037491ZpGAL4DBD in attP2) | *Shuai et al., 2023*; This paper | N.A. | Available from Aso lab |
| Strain, strain background (*Drosophila melanogaster*) | SS32189 (VT033047-p65ADZp in attP40 and R32D10ZpGAL4DBD in attP2) | *Shuai et al., 2023*; This paper | N.A. | Available from Aso lab |
| Strain, strain background (*Drosophila melanogaster*) | SS32218 (VT 040712-p65ADZp in attP40 and R13D05ZpGAL4DBD in attP2) | *Shuai et al., 2023*; This paper | N.A. | Available from Aso lab |
| Strain, strain background (*Drosophila melanogaster*) | SS32219 (VT 013618-p65ADZp in attP40 and VT 063636ZpGAL4DBD in attP2) | *Shuai et al., 2023*; This paper | N.A. | Available from Aso lab |
| Strain, strain background (*Drosophila melanogaster*) | SS32228 (VT 040004-p65ADZp in attP40 and VT 020600ZpGAL4DBD in attP2) | *Shuai et al., 2023*; This paper | N.A. | Available from Aso lab |

*Appendix 1 Continued on next page*

*Appendix 1 Continued*

| Reagent type (species) or resource | Designation | Source or reference | Identifiers | Additional information |
|---|---|---|---|---|
| Strain, strain background (*Drosophila melanogaster*) | SS32230 (VT 029362-p65ADZp in attP40 and VT013618ZpGAL4DBD in attP2) | *Shuai et al., 2023*; This paper | N.A. | Available from Aso lab |
| Strain, strain background (*Drosophila melanogaster*) | SS32244 (R13F04-p65ADZp in attP40 and R20H08ZpGAL4DBD in attP2) | *Shuai et al., 2023*; This paper | N.A. | Available from Aso lab |
| Strain, strain background (*Drosophila melanogaster*) | SS32254 (R26H05-p65ADZp in attP40 and VT 049923ZpGAL4DBD in attP2) | *Shuai et al., 2023*; This paper | N.A. | Available from Aso lab |
| Strain, strain background (*Drosophila melanogaster*) | SS33909 (VT 026342-p65ADZp in attP40 and VT 033912ZpGAL4DBD in attP2) | *Shuai et al., 2023*; This paper | N.A. | Available from Aso lab |
| Strain, strain background (*Drosophila melanogaster*) | SS33917 (VT 007746-p65ADZp in attP40 and R64A11ZpGAL4DBD in attP2) | *Yamada et al., 2023*; doi: https://doi.org/10.7554/eLife.79042 | N.A. | Available from Aso lab |
| Strain, strain background (*Drosophila melanogaster*) | SS33918 (VT 007746-p65ADZp in attP40 and R66B12ZpGAL4DBD in attP2) | *Yamada et al., 2023*; doi: https://doi.org/10.7554/eLife.79042 | N.A. | Available from Aso lab |
| Strain, strain background (*Drosophila melanogaster*) | SS39541 (R84C10-p65ADZp in attP40 and R23E10ZpGAL4DBD in attP2) | *Shuai et al., 2023*; This paper | N.A. | Available from Aso lab |
| Strain, strain background (*Drosophila melanogaster*) | SS40549 (R23E10-p65ADZp in JK73A and R84C10ZpGAL4DBD in attP2) | *Shuai et al., 2023*; This paper | N.A. | Available from Aso lab |
| Strain, strain background (*Drosophila melanogaster*) | SS41731 (R23E10-p65ADZp in JK22C and R84C10ZpGAL4DBD in attP2) | *Shuai et al., 2023*; This paper | N.A. | Available from Aso lab |
| Strain, strain background (*Drosophila melanogaster*) | SS45222 (VT018689-p65ADZp in attP40 and VT048933ZpGAL4DBD in attP2) | *Yamada et al., 2023*; doi: https://doi.org/10.7554/eLife.79042 | N.A. | Available from Aso lab |
| Strain, strain background (*Drosophila melanogaster*) | SS45234 (VT 026646-p65ADZp in attP40 and VT 029309ZpGAL4DBD in attP2) | *Yamada et al., 2023*; doi: https://doi.org/10.7554/eLife.79042 | N.A. | Available from Aso lab |
| Strain, strain background (*Drosophila melanogaster*) | SS48882 (R54H04-p65ADZp in attP40 and R26C06ZpGAL4DBD in attP2) | *Shuai et al., 2023*; This paper | N.A. | Available from Aso lab |
| Strain, strain background (*Drosophila melanogaster*) | SS48899 (R89B06-p65ADZp in attP40 and VT 063627ZpGAL4DBD in attP2) | *Shuai et al., 2023*; This paper | N.A. | Available from Aso lab |
| Strain, strain background (*Drosophila melanogaster*) | SS48900 (R91F05-p65ADZp in attP40 and VT 054914ZpGAL4DBD in attP2) | *Shuai et al., 2023*; This paper | N.A. | Available from Aso lab |
| Strain, strain background (*Drosophila melanogaster*) | SS49755 (R56B05-p65ADZp in attP40 and R84B09ZpGAL4DBD in attP2) | *Shuai et al., 2023*; This paper | N.A. | Available from Aso lab |
| Strain, strain background (*Drosophila melanogaster*) | SS49897 (R26C06-p65ADZp in VK00027 and R54H04ZpGAL4DBD in attP2) | *Shuai et al., 2023*; This paper | N.A. | Available from Aso lab |
| Strain, strain background (*Drosophila melanogaster*) | SS49899 (R26C06-p65ADZp in su(Hw)attP8 and R54H04ZpGAL4DBD in attP2) | *Shuai et al., 2023*; This paper | N.A. | Available from Aso lab |
| Strain, strain background (*Drosophila melanogaster*) | SS56699 (TH-p65ADZp in VK00027 and VT025720ZpGAL4DBD in attP2) | *Hulse et al., 2021*; https://doi.org/10.7554/eLife.66039 | N.A. | |
| Strain, strain background (*Drosophila melanogaster*) | SS67221 (VT 026646-p65ADZp in attP40 and VT 019911ZpGAL4DBD in attP2) | *Yamada et al., 2023*; doi: https://doi.org/10.7554/eLife.79042 | N.A. | Available from Aso lab |
| Strain, strain background (*Drosophila melanogaster*) | SS88953 (VT014604-p65ADZp in attP40 and VT063740ZpGAL4DBD in attP2) | *Shuai et al., 2023*; This paper | N.A. | Available from Aso lab |
| Strain, strain background (*Drosophila melanogaster*) | SS88997 (R25G01-p65ADZp in JK22C and R15D05ZpGAL4DBD in attP2) | *Shuai et al., 2023*; This paper | N.A. | Available from Aso lab |
| Strain, strain background (*Drosophila melanogaster*) | pJFRC100-20XUAS-TTS-Shibire-ts1-p10 in VK00005 | *Pfeiffer et al., 2012*; https://doi.org/10.1073/pnas.120452010 | N.A. | |
| Strain, strain background (*Drosophila melanogaster*) | UAS-TeNT | *Keller et al., 2002*: PMID:11810637 | N.A. | |
| Antibody | Anti-GFP (rabbit polyclonal) | Invitrogen | A11122 RRID:AB_221569 | 1:1000 |
| Antibody | Anti-Brp (mouse monoclonal) | Developmental Studies Hybridoma Bank | nc82 RRID:AB_2341866 | 1:30 |
| Antibody | Anti-HA-Tag (mouse monoclonal) | Cell Signaling Technology | C29F4; #3724 RRID:AB_10693385 | 1:300 |
| Antibody | Anti-FLAG (rat monoclonal) | Novus Biologicals | NBP1-06712 RRID:AB_1625981 | 1:200 |
| Antibody | Anti-V5-TAG Dylight-549 (mouse monoclonal) | Bio-Rad | MCA2894D549GA RRID:AB_10845946 | 1:500 |
| Antibody | Anti-mous IgG(H&L) AlexaFluor-568 (goat polyclonal) | Invitrogen | A11031 RRID:AB_144696 | 1:400 |
| Antibody | Anti-rabbit IgG(H&L) AlexaFluor-488 (goat polyclonal) | Invitrogen | A11034 RRID:AB_2576217 | 1:800 |

*Appendix 1 Continued*

| Reagent type (species) or resource | Designation | Source or reference | Identifiers | Additional information |
|---|---|---|---|---|
| Antibody | Anti-mouse IgG(H&L) AlexaFluor-488 conjugated (donkey polyclonal) | Jackson Immuno Research Labs | 715-545-151 RRID:AB_2341099 | 1:400 |
| Antibody | Anti-rabbit IgG(H&L) AlexaFluor-594 (donkey polyclonal) | Jackson Immuno Research Labs | 711-585-152 RRID:AB_2340621 | 1:500 |
| Antibody | Anti-rat IgG(H&L) AlexaFluor-647 (donkey polyclonal) | Jackson Immuno Research Labs | 712-605-153 RRID:AB_2340694 | 1:300 |
| Antibody | Anti-Mouse IgG (H&L) ATTO 647N (goat polyclonal) | ROCKLAND | 610-156-121 RRID:AB_10894200 | 1:100 |
| Antibody | Anti-rabbit IgG (H+L) Alexa Fluor 568 (goat polyclonal) | Invitrogen | A-11036 RRID:AB_10563566 | 1:1000 |
| Chemical compound, drug | 4-Methylcyclohexanol | VWR | AAA16734-AD | |
| Chemical compound, drug | Pentyl acetate | Sigma-Aldrich | 109584 | |
| Chemical compound, drug | Ethyl lactate | Sigma-Aldrich | W244015 | |
| Chemical compound, drug | Paraffin oil | Sigma-Aldrich | 18512 | |
| Software, algorithm | ImageJ and Fiji | *Schneider et al., 2012* | https://imagej.nih.gov/ij/http://fiji.sc/ | |
| Software, algorithm | MATLAB | MathWorks | https://www.mathworks.com/ | |
| Software, algorithm | Adobe Illustrator CC | Adobe Systems | https://www.adobe.com/products/illustrator.html | |
| Software, algorithm | GraphPad Prism 9 | GraphPad Software | https://www.graphpad.com/scientific-software/prism/ | |
| Software, algorithm | Python | Python Software Foundation | https://www.python.org/ | |
| Software, algorithm | neuPrint | HHMI Janelia | https://doi.org/10.25378/janelia.12818645.v1 | |
| Software, algorithm | Cytoscape | *Shannon et al., 2003* | https://cytoscape.org/ | |
| Software, algorithm | NeuTu | *Zhao et al., 2018* | https://github.com/janelia-flyem/NeuTu; *janelia-flyem, 2018* | |
| Software, algorithm | ScanImage | Vidrio Technologies | https://vidriotechnologies.com/ | |
| Software, algorithm | VVDveiwer | HHMI Janelia | https://github.com/takashi310/VVD_Viewer | |
| Other | Grade 3MM Chr Blotting Paper | Whatmann | 3030-335 | Used in glass vials with paraffin oil diluted odors |
| Other | Mass flow controller | Alicat | MCW-200SCCM-D | Mass flow controller used for the olfactory arena |

