## [Editor Report]

This study provides important new insights into how learning affects behavior in the *Drosophila* model. Using a combination of connectomics, neurophysiology, and behavioral analysis, a small group of neurons in the *Drosophila* brain that integrates learned odor valences and promotes odor tracking by driving upwind orientation and movement is described. The study's conclusion is supported by convincing evidence and rigorous quantitative analysis. Insights from the neural circuit mechanism that translates learning-induced plasticity into appropriate behavioral actions will be of broad interest to neuroscientists.

---

## [Decision Letter]

**Decision letter after peer review:**

Thank you for submitting your article "Neural circuit mechanisms for transforming learned olfactory valences into wind-oriented movement" for consideration by *eLife*. Your article has been reviewed by 3 peer reviewers, and the evaluation has been overseen by a Reviewing Editor and K VijayRaghavan as the Senior Editor. The following individual involved in the review of your submission has agreed to reveal their identity: Suewei Lin (Reviewer #2).

Essential revisions:

As you will see from the comments, the reviewers find your study very interesting and important. While several reviewers have raised the possibility that additional experiments are necessary and have suggested those accordingly in their comments, we have jointly decided that the current experimental evidence is sufficient, provided that you address all comments in full by making substantial changes to the text, figures, and legends. For instance, the specificity of the role of the UpWin neurons in this particular form of learning and behavior was criticised in various ways by the reviewers.

*Reviewer #1 (Recommendations for the authors):*

The presented work represents an important contribution to the question of how memory retrieval is translated into behavior by circuits downstream of the mushroom body. However, I find some of the conclusions require additional experimental support detailed in the points below.

1. The authors show in Figure 1 that wind orientation is not a general feature of appetitive memory recall, but specific to memories encoded in the A1 compartment of the MB. Memory in this compartment is supposed to build up slowly and support longer-lasting memories specifically. The authors state themselves "DANs in the α1 only weakly respond to sugar (Yamagata et al., 2015), and write a memory slowly even when optogenetically activated (Aso and Rubin, 2016; Yamada et al., unpublished)", Discussion line 470. They conclude that "these features collectively indicate that flies undergo wind-guided olfactory navigation only when they expect a robust reward associated with the odor" (Discussion line 474). However, the behavior they observe upon artificial training with α1 DANs in Figure 1 is measured immediately after training, as is the change in odor response by the appetitively associated odor in Figure 5 (timepoint of test measurement should be indicated in the figure). Therefore, it rather seems that upwind locomotion is present at early timepoints, and plasticity changes in that compartment are not restricted to slow processes leading exclusively to LTM. The authors should either provide more experimental data that support their hypothesis or tune down their claims and discuss in more detail how they view the involvement of α1 in short- and long-term processing of memory.

2. The authors state several times that the identified UPWINs downstream of the MB can decode opposing values. This is based on the assumption that these neurons get excitatory and inhibitory input from different MBONs A1 and A3 involved in appetitive and aversive memory, respectively. However, their approach is solely based on appetitive data. Moreover, they only show that activation of A3 is leading to activation of UPWINs in physiological assays (Figure 4). In fact, their conclusion that increased UPWIN activity after appetitive memory formation results from disinhibition by decreasing activity in A1 (Figure 5, Results line 276) is pure speculation since they never show that inhibition of A1 is leading to increased UPWIN activity. It seems more likely to me that appetitive memory is encoded by increased A3 as well as decreased A1 activity, which would lead to an additive effect towards UPWIN activation. Increased A3 activity has been repeatedly shown to result from appetitive training, as the authors state themselves (Results line 211). The authors should prove opposing effects for aversive memory recall or solely focus the conclusions on appetitive memories. Also, optogenetic inhibition of A1 using e.g. UAS-GTCR in electrophysiology and/or calcium imaging is highly recommended to back up their conclusion about UPWIN disinhibition. Taken together, the identified circuit motive could well support the valence balance model of the MB and the encoding of parallel memories, aversive and appetitive, but this is not shown by the presented data.

3. The function of the UPWINs seems complex and go beyond upwind orientation. The data convincingly show that UPWINs promote wind-oriented behavior. However, the fact that flies need to be starved during opto-induced upwind movement as well as during the memory test, raises the question of how these neurons are involved in naïve state-dependent odor behaviors. Starved flies follow odor plumes even in the absence of a memory, so it would be interesting how such behavior is altered when blocking UPWINs. The authors describe the very interesting effect that activation of UPWINs is leading to a significant returning behavior to the place of activation. In the discussion, the authors state: In addition to promoting an upwind surge, UpWiNs activation increased the probability of returning to the location where activation was applied even after the cessation of both optogenetic activation and airflow. Therefore, UpWiNs alone may be able to promote a series of cast-surge-cast reactions when flies navigate intermittent plumes of reward-predicting odors (line 409 continuing). However, the experimental design in Figure 2 does not include odors. The fact that UPWINs get input from Lateral Horn neurons and involve neurons of the Fan-shaped body, it seems more likely that UPWINs function as a more general context-dependent integrator of spatial cues and odors. While I appreciate that this is out of the scope of this study, it raises critical questions about the conclusions drawn here. I would suggest the authors either keep the returning data for a follow-up story or provide more detailed information about possible UPWIN functions here.

*Reviewer #2 (Recommendations for the authors):*

Here are some suggestions that I believe would improve the clarity and readability of the manuscript.

1. L120-L121: References are needed for the statement that MBON-α1 and MBON-α3 convey long-term appetitive and aversive memories, respectively.

2. It would be helpful to include empty GAL4 traces in Figures 1E and 1G.

3. The behavioral analysis in Figure 2 requires additional clarification. The sample size of n = 18-132 is a wide range, and it would be useful to provide the sample size for each GAL4 line. Furthermore, it is not clear from the figure whether each data point represents an average of the six LED exposures or whether each exposure contributes to a data point. Additionally, the schematic in Figure 2A, which depicts the revisit probability measurement, may not be clear to some readers. To improve the clarity of this figure, the authors may consider using the schematic in Figure 7—figure supplement 1 and providing a further explanation in the figure legend.

4. I am confused by the CRE39 results in Figure 2—figure supplement 1. These flies show negative displacement while orienting more toward the upwind direction during LED exposure. Why is this?

5. It would be helpful to have synaptotagmin-smGFP-v5 staining for SS33917 and SS33918, similar to the staining shown for some other lines in Figure 2—figure supplements 3 and 4.

6. Figure 2—figure supplement 4 is not mentioned in the text.

7. The section at lines 234-240 may be more appropriately placed in the Discussion section.

8. It would be helpful to show the expression pattern of R64A11-LexA.

---

## [Author Response]

Essential revisions:As you will see from the comments, the reviewers find your study very interesting and important. While several reviewers have raised the possibility that additional experiments are necessary and have suggested those accordingly in their comments, we have jointly decided that the current experimental evidence is sufficient, provided that you address all comments in full by making substantial changes to the text, figures, and legends. For instance, the specificity of the role of the UpWin neurons in this particular form of learning and behavior was criticised in various ways by the reviewers.

We would like to thank reviewers for constructive comments to improve the manuscript. We are pleased to know that they agreed that no additional experiments are required for publication. We nonetheless added some new data to strengthen our conclusions in this revision and adjusted the text and figures as detailed in the responses to the comments below.

Reviewer #1 (Recommendations for the authors):The presented work represents an important contribution to the question of how memory retrieval is translated into behavior by circuits downstream of the mushroom body. However, I find some of the conclusions require additional experimental support detailed in the points below.1. The authors show in Figure 1 that wind orientation is not a general feature of appetitive memory recall, but specific to memories encoded in the A1 compartment of the MB. Memory in this compartment is supposed to build up slowly and support longer-lasting memories specifically. The authors state themselves "DANs in the α1 only weakly respond to sugar (Yamagata et al., 2015), and write a memory slowly even when optogenetically activated (Aso and Rubin, 2016; Yamada et al., unpublished)", Discussion line 470. They conclude that "these features collectively indicate that flies undergo wind-guided olfactory navigation only when they expect a robust reward associated with the odor" (Discussion line 474). However, the behavior they observe upon artificial training with α1 DANs in Figure 1 is measured immediately after training, as is the change in odor response by the appetitively associated odor in Figure 5 (timepoint of test measurement should be indicated in the figure). Therefore, it rather seems that upwind locomotion is present at early timepoints, and plasticity changes in that compartment are not restricted to slow processes leading exclusively to LTM. The authors should either provide more experimental data that support their hypothesis or tune down their claims and discuss in more detail how they view the involvement of α1 in short- and long-term processing of memory.

We did not intend to claim that wind-directional response to CS+ odor is exclusively for LTM. Blocking DANs in the alpha1 compartment does not impair the memory measured by T-maze immediately after odor-sugar conditioning (Yamagata et al., 2015). However, it does not necessarily indicate the lack of plasticity in the alpha1 compartment immediately after training; it may simply mean that short-term memories in other compartments can drive conditioned behaviors. When optogenetic activation of DANs in the alpha1 compartment was used as US, flies showed conditioned behaviors and MBON-alpha1 reduced response to CS+ odor immediately after training (Figure 1) (Aso and Rubin, 2016; Yamada et al., 2023). Compared to DANs in other compartments, longer/repetitive training was required for fully inducing memory with optogenetic activation of DANs in the alpha1 compartment, even though they showed similar response properties to release dopamine (Yamada et al., 2023). Thus, we expect that plasticity in alpha1 can be present immediately after training but its magnitude tends to be weaker than that in other compartments when training is short and not repeated. If this view is correct, alpha1-specific conditioned behavior (i.e. upwind response) should need more training to develop compared to conditioned preference between CS+/CS- odors.

To clarify this point, in Figure 1C-F, we have added new data to compare the rates to develop the learned odor preference in binary choice and wind-directional response to CS+ when flies are repeatedly trained by pairing an odor with optogenetic activation of sugar sensory neurons, not DANs. This data indicates that more training is required to form memories to drive upwind response to CS+ odor. After single training, flies already showed strong preference to CS+ odor over CS- odor (PI=0.56) but did not show significant upwind response to CS+ odor. Therefore, these observations are consistent with our expectation based on the results of direct DAN activations: Short single training with optogenetic activation of PAM-alpha1 DANs failed to induce memories, whereas other DANs in gamma4 and gamma5 could induce significant memories (Yamada et al., 2023).

We would also like to point out that this concern does not affect our main conclusion. Through its versatile functions, UpWiNs have a potential to contribute to promotion of navigational behaviors whenever activated with natural stimuli irrespective of underlying anatomical pathways. Whether one can detect the requirement of UpWiNs in a specific form of learning or memory phases would depend on the relative degree of plasticity in each MB compartments, their functional redundancy and existence of connections between UpWiNs and MBONs. Based on the anatomical knowledge (direct connection with MBON-alpha1) and long retention of memory in the α 1 compartment, we tested the effect of blocking UpWiNs one-day after odor-sugar conditioning (Figure 7A-B). However, requirement in other memory phases and innate odor requires further investigations. Among hundreds of cell types that are immediate downstream of MBONs (Li et al. 2020), our study identified the first cell type that acquires enhanced odor response to reward-predicting odor, can promote wind-directional turning and is required when tested one-day after odor-sugar conditioning.

The time point of the test measurement is now indicated in Figure 5.

2. The authors state several times that the identified UPWINs downstream of the MB can decode opposing values. This is based on the assumption that these neurons get excitatory and inhibitory input from different MBONs A1 and A3 involved in appetitive and aversive memory, respectively. However, their approach is solely based on appetitive data. Moreover, they only show that activation of A3 is leading to activation of UPWINs in physiological assays (Figure 4). In fact, their conclusion that increased UPWIN activity after appetitive memory formation results from disinhibition by decreasing activity in A1 (Figure 5, Results line 276) is pure speculation since they never show that inhibition of A1 is leading to increased UPWIN activity. It seems more likely to me that appetitive memory is encoded by increased A3 as well as decreased A1 activity, which would lead to an additive effect towards UPWIN activation. Increased A3 activity has been repeatedly shown to result from appetitive training, as the authors state themselves (Results line 211). The authors should prove opposing effects for aversive memory recall or solely focus the conclusions on appetitive memories. Also, optogenetic inhibition of A1 using e.g. UAS-GTCR in electrophysiology and/or calcium imaging is highly recommended to back up their conclusion about UPWIN disinhibition. Taken together, the identified circuit motive could well support the valence balance model of the MB and the encoding of parallel memories, aversive and appetitive, but this is not shown by the presented data.

We did not intend to exclude the possibility suggested by the Reviewer. The logic underlying the experimental design in Figure 4C was as follows. In naïve flies, both MBON-α1 and MBON-α3 respond to odors (Hige et al., 2015). In that sense, simultaneous activation of MBON-α1 and MBON-α3 mimics a situation in naïve flies. After training and induction of synaptic depression in α1, MBON-α1 reduces the odor response (Yamada et al., 2023), whereas MBON-α3 enhances odor response (Plaçais et al., 2013) presumably due to intercompartmental connections (Takemura et al., 2017; Tanaka et al., 2008). Therefore, MBON-α3 alone activation in Figure 4C mimics a situation after appetitive odor conditioning. As shown in Figure 3C, UpWiNs collectively receive inputs from excitatory and inhibitory cell types including the lateral horn neurons. Any skewed inputs would activate UpWiNs, but MBON-α1 and MBON-α3 are predominant inputs. In response to the reviewer’s comment, we removed the term “disinhibition” from our Summary and adjusted the text to convey this view.

3. The function of the UPWINs seems complex and go beyond upwind orientation. The data convincingly show that UPWINs promote wind-oriented behavior. However, the fact that flies need to be starved during opto-induced upwind movement as well as during the memory test, raises the question of how these neurons are involved in naïve state-dependent odor behaviors. Starved flies follow odor plumes even in the absence of a memory, so it would be interesting how such behavior is altered when blocking UPWINs.

We suspect that UpWiNs may respond to food-related odors in naïve flies and mediate both innate and learned wind-directional behaviors for the reasons mentioned in Discussion. EM connectome analysis revealed direct connections from the lateral horn neurons to UpWiNs (Figure 3C). We also noted in Discussion that the pattern of dopamine release in response to vinegar (i.e. upregulation in gamma4, gamma5 and downregulation in gamma2) is similar to the pattern of dopamine release by UpWiN activation (Yamada et al. 2023).

“Interestingly, the patterns of DAN population responses to SMP108 or UpWiNs activation are similar to those observed when flies are walking toward vinegar in a virtual environment (Zolin et al., 2021). Together with the evidence of inputs from the lateral horn neurons, this may indicate that UpWiNs is also responsible for upwind locomotion to innately attractive odors and can be the causal source of action correlates in DANs.”

These data may indicate that UpWiNs are involved in wind-directional response to innately attractive odors.

The authors describe the very interesting effect that activation of UPWINs is leading to a significant returning behavior to the place of activation. In the discussion, the authors state: In addition to promoting an upwind surge, UpWiNs activation increased the probability of returning to the location where activation was applied even after the cessation of both optogenetic activation and airflow. Therefore, UpWiNs alone may be able to promote a series of cast-surge-cast reactions when flies navigate intermittent plumes of reward-predicting odors (line 409 continuing). However, the experimental design in Figure 2 does not include odors.

This is a speculative statement in Discussion to put our observation in optogenetic experiments in a context of olfactory navigation literature and suggest a hypothetical role of UpWiNs in response to intermittent odor. Our assumption is that the UpWiNs activity would be terminated when flies leave a plume of reward-predicting odor, and that termination of optogenetic activation of UpWiNs is a similar experience for flies. If necessary and recommended by reviewers and editors, we can place this section under “Ideas and Speculations”.

The fact that UPWINs get input from Lateral Horn neurons and involve neurons of the Fan-shaped body, it seems more likely that UPWINs function as a more general context-dependent integrator of spatial cues and odors. While I appreciate that this is out of the scope of this study, it raises critical questions about the conclusions drawn here. I would suggest the authors either keep the returning data for a follow-up story or provide more detailed information about possible UPWIN functions here.

In addition to the upwind steering, optogenetic activation UpWiNs can promote other kinds of behaviors even in the absence of airflow (Figure 7E and Figure 7 figure-supplement 1). In the recent paper (Yamada et al. 2023), we also showed that UpWiNs can promote release of dopamine in multiple compartments and suggested its role in second-order conditioning. Therefore, the function of UpWiNs is not restricted to promotion of upwind locomotion. We believe the versatile functions of UpWiNs including returning phenotype and excitatory drive to DANs make these neurons interesting and special. Therefore, we would like to keep the returning phenotype data.

Reviewer #2 (Recommendations for the authors):Here are some suggestions that I believe would improve the clarity and readability of the manuscript.1. L120-L121: References are needed for the statement that MBON-α1 and MBON-α3 convey long-term appetitive and aversive memories, respectively.

We rephrased the sentence and added citations.

“UpWiNs receive inputs from several types of lateral horn neurons and integrate inhibitory and excitatory inputs from MBON-α1 and MBON-α3, which are the output neurons of MB compartments that store long-lasting appetitive or aversive memories, respectively (Aso and Rubin, 2016; Ichinose et al., 2015; Jacob and Waddell, 2022; Pai et al., 2013; Yamagata et al., 2015).”

2. It would be helpful to include empty GAL4 traces in Figures 1E and 1G.

We did not include empty-split-GAL4 traces in Figure 1E and 1G because the space is limited and the key data are shown in Figure F and H. Also, empty-split-GAL4>CsChrimson flies did not change odor response by odor-LED training as shown in Figure 1—figure supplement 1.

3. The behavioral analysis in Figure 2 requires additional clarification. The sample size of n = 18-132 is a wide range, and it would be useful to provide the sample size for each GAL4 line.

The screening was performed in multiple batches and some lines such as Empty-split GAL4 and MB011C served as controls and tested across batches. The sample size and numerical values are in Supplementary File 2 source data files.

Furthermore, it is not clear from the figure whether each data point represents an average of the six LED exposures or whether each exposure contributes to a data point.

The dots in the box plots data points outside the quartile range and represent data from each of 6 movies. The conclusions about the UpWiNs lines (i.e. SS33917 and SS33918) did not change when trial averages of 6 movies were used for statistical tests.

Additionally, the schematic in Figure 2A, which depicts the revisit probability measurement, may not be clear to some readers. To improve the clarity of this figure, the authors may consider using the schematic in Figure 7—figure supplement 1 and providing a further explanation in the figure legend.

We’ve revised the schematic in Figure 2A and legend.

4. I am confused by the CRE39 results in Figure 2—figure supplement 1. These flies show negative displacement while orienting more toward the upwind direction during LED exposure. Why is this?

As the reviewer pointed out, SS32218>CsChrimson flies increased walking speed but their mean orientation toward upwind was indistinguishable compared to the empty-split GAL4 control (Figure 2—figure supplement 1). It is possible that flies preferentially increase forward walking speed when they are facing downwind, but we will need more trajectories to perform such analysis. The negative Z-score is relative to other lines. Flies tend to move slightly toward the edges of the arena under intense red light rather than upwind response, possibly due to thigmotaxis. Optogenetic activation of CRE39 and/or off-targeted neurons in SS33218 may prevent this basal response to LED light rather than promoting downwind walking.

5. It would be helpful to have synaptotagmin-smGFP-v5 staining for SS33917 and SS33918, similar to the staining shown for some other lines in Figure 2—figure supplements 3 and 4.

We added an image as an insert in Figure 3A.

6. Figure 2—figure supplement 4 is not mentioned in the text.

We corrected the figure citations to mention this figure.

“These behavioral data can be immediately put into the context of the EM connectome map, since the cell types in each driver lines were morphologically matched by comparing confocal and electron microscope images (see examples in Figure 2—figure supplement 3 and 4). “

7. The section at lines 234-240 may be more appropriately placed in the Discussion section.

We would like to keep these sentences in the result, since convergence to the SMP108 was a feature common to UpWiNs and readers would need explanation about what SMP108 is.

8. It would be helpful to show the expression pattern of R64A11-LexA.

The expression patterns of R64A11-LexA in attP40 are available on the website of the Janelia Flylight team.

https://flweb.janelia.org/cgi-bin/view_flew_imagery.cgi?line=R64A11L